# Cytokinetic abscission requires actin-dependent microtubule severing

Tamara Advedissian ●[1], Stéphane Frémont[1] & Arnaud Echard ●[1] ✉

Cell division is completed by the abscission of the intercellular bridge connecting the daughter cells. Abscission requires the polymerization of an ESCRT-III cone close to the midbody to both recruit the microtubule severing enzyme spastin and scission the plasma membrane. Here, we found that the microtubule and the membrane cuts are two separate events that are regulated differently. Using HeLa cells, we uncovered that the F-actin disassembling protein Cofilin-1 controls the disappearance of a transient pool of branched F-actin which is precisely assembled at the tip of the ESCRT-III cone shortly before the microtubule cut. Functionally, Cofilin-1 and Arp2/3-mediated branched F-actin favor abscission by promoting local severing of the microtubules but do not participate later in the membrane scission event. Mechanistically, we propose that branched F-actin functions as a physical barrier that limits ESCRT-III cone elongation and thereby favors stable spastin recruitment. Our work thus reveals that F-actin controls the timely and local disassembly of microtubules required for cytokinetic abscission.

Cytokinesis concludes cell division and leads to the physical separation of the two daughter cells following chromosome segregation. After complete furrow ingression driven by an actomyosin ring[1,2], the daughter cells are connected by an intercellular bridge (ICB) filled with microtubules (MTs). At the center of the ICB, an electron-dense structure named the midbody is a major platform which recruits all known proteins required for cell separation also called abscission[3–5]. In particular, the ESCRT-III (Endosomal Sorting Complex Required for Transport III) machinery[6–10] localizes at the midbody, and highly dynamic ESCRT-III filaments polymerize as a cone-like structure extending from the midbody to constrict the ICB membrane until the final cut[11–17]. This site of constriction has been named the "secondary ingression" or "abscission site" or "constriction zone" and is usually located at a distance of 1-2 μm from the midbody[18–22]. Scission of the ICB often occurs successively on both sides of the midbody, releasing a "midbody remnant" that can be later tethered and phagocytosed[23,24].

Cytokinetic abscission requires a profound remodeling of the cytoskeleton at the ICB. Notably, both MTs and actin filaments (F-actin) must be removed to allow the ICB cut since the presence of these cytoskeleton elements would prevent local membrane fusion[16]. In addition to mediating membrane scission, the ESCRT-III machinery

contributes to the clearance of MTs at the secondary ingression by recruiting the MT severing AAA-ATPase spastin through direct binding with the ESCRT-III subunit CHMP1B[13,25–27]. In parallel, several pathways control global F-actin levels within the ICB by either limiting F-actin polymerization −through RhoA inactivation by p50RhoGAP or PtdIns(4,5)P2 hydrolysis by OCRL1− or triggering F-actin depolymerization −through actin oxidation by MICAL1[19,28–32]. While many studies demonstrated that global F-actin disassembly within the ICB is critical for abscission, it remains unclear whether F-actin might also play, likely locally, a role in abscission. Indeed, depending on the studies, experimental depolymerization of F-actin after complete furrow ingression using drugs led either to no effect, or to ICB instability followed by the formation of binucleated cells or to abscission defects[13,14,33,34]. Importantly, if F-actin is required for abscission, it remains to be discovered why at the mechanistic level.

Here, we found that the F-actin disassembling protein Cofilin-1 participates in cytokinetic abscission by regulating a transient pool of Arp2/3-mediated branched F-actin at the secondary ingression. These findings thus reveal an undescribed function of Cofilin-1 in late steps of cytokinesis, long after furrow ingression during which Cofilin-1 is well known to favor F-actin turnover[35–41]. Cofilin-1 and Arp2/3 promote

[1]Institut Pasteur, Université Paris Cité, CNRS UMR3691, Membrane Traffic and Cell Division Unit, 25-28 rue du Dr Roux, F-75015 Paris, France.
✉e-mail: arnaud.echard@pasteur.fr

abscission by controlling the severing of the MTs but not the membrane scission that we found to occur much later. Thus, abscission requires two temporally distinct events, a MT cut and a membrane cut, that are regulated separately. Furthermore, our work reveals a critical and specific role of F-actin in the ESCRT-III- and spastin-dependent MT severing step required for abscission.

## Results

### Cofilin-1 and F-actin are transiently enriched at the secondary ingression

We recently obtained the quantitative proteome of midbody remnants from HeLa cells, that we termed the Flemmingsome[21]. This helped to reveal the function of several proteins in abscission (e.g. Syndecan-4, syntenin, Cavin1, Caveolin-1)[21,22] and post-abscission midbody remnant tethering to cells (e.g., BST2/Tetherin)[42]. Here, we noticed that Cofilin-1 was found enriched by two-fold in the Flemmingsome compared to control fractions, which suggested a potential function of this actin binding protein in late cytokinetic steps (Supplementary Fig. 1a).

We first investigated the localization of endogenous Cofilin-1 throughout cytokinesis by immunofluorescence in HeLa cells. As previously described[43], Cofilin-1 was detected at the cleavage furrow in $71.8\% \pm 3.4\%$ of the dividing cells ($N = 4$ independent experiments, $n = 15–43$ cells per experiment) (Supplementary Fig. 1b). A thus far overlooked pool of Cofilin-1 was also present in the ICBs at later stages of cytokinesis. Cofilin-1 indeed localized at the midbody in $60.6\% \pm 5.4\%$ ($N = 4$, $n = 22–42$) of late ICBs without yet secondary ingression (i.e. with no pinch of the tubulin staining on the side of the midbody) (Fig. 1a, top panels). Furthermore, in $71.2\% \pm 7.9\%$ ($N = 4$, $n = 15–19$) of older ICBs with secondary ingression, Cofilin-1 was detected at the midbody and/or, intriguingly, at the secondary ingression itself (Fig. 1a bottom panels, arrowhead). The latter localization was observed in approximately 30% of ICBs with secondary ingression (Fig. 1b), suggesting that Cofilin-1 is transiently recruited at the secondary ingression. We confirmed the localization of Cofilin-1 at the secondary ingression in primary BMEL (Bipotential Mouse Embryonic Liver) progenitor cells (Supplementary Fig. 1c).

To reveal the dynamic behavior of Cofilin-1 in late cytokinetic steps, we monitored HeLa cells that expressed wild type or mutant versions of human Cofilin-1 tagged with GFP (Cofilin-1-GFP) by time-lapse spinning disk microscopy. We observed that both wild type and active, non-phosphorylatable Cofilin-1 mutant (Cofilin-1-S3A-GFP), exhibited a transient recruitment at the secondary ingression (Fig. 1c arrowhead and Supplementary Fig. 1d). In contrast, the phosphomimetic mutant (Cofilin-1-S3E-GFP), which is inactive and had a lower affinity for actin[44–47], did not localize to ICBs (Supplementary Fig. 1e). This result suggested that Cofilin-1 recruitment at the secondary ingression was dependent on the binding of activated Cofilin-1 to F-actin. Consistently, live cell microscopy in cells that stably co-expressed Cofilin-1-GFP and the F-actin probe LifeAct-mCherry showed that Cofilin-1 and F-actin appeared and disappeared simultaneously at the secondary ingression (Fig. 1d, arrowhead and Supplementary movie 1). In addition, treating the cells with high doses of either Cytochalasin D (CytoD) or Latrunculin A (LatA) −two drugs that depolymerize F-actin through different mechanisms− reduced F-actin levels in the ICBs and the proportion of ICBs with Cofilin-1 at the secondary ingression (Fig. 1e and Supplementary Fig. 1f, g). We conclude that Cofilin-1 and F-actin are transiently enriched together at the secondary ingression and that Cofilin-1 recruitment depends on F-actin.

### Cofilin-1 contributes to abscission by regulating F-actin depolymerization

To investigate a potential function of Cofilin-1 in abscission, we depleted Cofilin-1 using siRNAs (Fig. 2a) and used time-lapse phase contrast microscopy to measure the time it took after furrow ingression to physically cut the ICB. In our depletion conditions,

approximately 22% of the dividing cells became binucleated (Supplementary Fig. 2a). Consistent with previous reports[38,48], a fraction of binucleated cells arose before ICB formation either from instability of the ingressing furrow or from mispositioning of the contractile ring (Supplementary Fig. 2b). We also observed that approximately 40% of the binucleated cells resulted from late regression of the ICB, a phenomenon not reported before. This often occurred at a time after normal abscission (i.e. > 250 min), indicating cytokinesis abortion consecutive to failed abscission upon Cofilin-1 depletion (Supplementary Fig. 2b). Importantly, abscission was delayed in Cofilin-1-depleted cells that did not become binucleated (Fig. 2b). Transient expression of wild type Cofilin-1-GFP rescued normal timing of abscission in Cofilin-1-depleted cells (Fig. 2c and Supplementary Fig. 2c), indicating a specific function of Cofilin-1 in abscission. Moreover, active Cofilin-1-S3A-GFP mutant restored correct timing of abscission whereas the inactive form, Cofilin-1-S3E-GFP, did not (Fig. 2c), showing that Cofilin-1 activation and binding to actin are required for Cofilin-1 function in abscission.

To rule out that the abscission delay resulted from indirect effects due to long term depletion of Cofilin-1 by siRNAs, we took advantage of a chemogenetic method recently developed[49] to acutely deplete Cofilin-1 from the cytosol only in cytokinesis and after furrow ingression. Briefly, we generated a stable cell line that expressed both siRNA-resistant Cofilin-1-GFP and engineered anti-GFP nanobodies anchored to the outer membrane of mitochondria (Fig. 2d). After depletion of endogenous Cofilin-1, removal of trimethoprim (TMP) from the medium activated the nanobodies and thus led to a rapid (<5 min) and massive sequestration of Cofilin-1-GFP from the cytosol to the mitochondrial surface (Fig. 2d), preventing Cofilin-1-GFP recruitment at the secondary ingression (Supplementary Fig. 2d). Interestingly, acute depletion of Cofilin-1-GFP from the cytosol, after furrow ingression, delayed abscission (Fig. 2e, siCofilin-1 - TMP) as compared to control conditions (Fig. 2e, siCofilin-1 + TMP and siControl ± TMP), similar to what we observed upon long-term depletion. Thus, acute depletion of Cofilin-1 demonstrated a function of Cofilin-1 after furrow ingression in promoting cytokinetic abscission.

To determine whether Cofilin-1 participates in abscission through depolymerization of F-actin, as expected for an actin disassembly factor, we treated the cells with low doses of either CytoD (Fig. 2f) or LatA (Supplementary Fig. 2e). The addition of low doses of either Cyto D (7.5 nM) or LatA (20 nM) did not perturb cytokinetic abscission in control cells but restored proper timing of abscission in Cofilin-1-depleted cells. We conclude that the disassembly of F-actin by Cofilin-1 is important for abscission.

### Cofilin-1 controls the local disassembly of branched F-actin at the secondary ingression just before the MT cut

We next investigated the nature of the pool of F-actin colocalizing with Cofilin-1 at the secondary ingression and whether Cofilin-1 controls its dynamics. In fixed HeLa cells, F-actin at the secondary ingression co-localized with the subunit ARPC2 of the Arp2/3 complex, a major nucleator of actin filaments, primarily in the form of branches growing off the sides of other filaments (branched F-actin)[50–54] (Fig. 3a, arrowhead). The presence of ARPC2 at the secondary ingression was also observed in mouse BMEL cells (Supplementary Fig. 3a). Time-lapse spinning disk microscopy in HeLa cells further established that the Arp2/3 subunit GFP-ARPC4 co-localized with Cofilin-1 at the secondary ingression (Fig. 3b and Supplementary movie 2). Of note, Cofilin-1 and Arp2/3 appeared simultaneously at the secondary ingression, displayed a maximal recruitment 5 min before the MT cut in most cells and completely disappeared when the MTs were cut (Fig. 3c, time 0 corresponds to the time of the MT cut to allow comparison between cells and Supplementary Fig. 3b). Both Cofilin-1 and ARPC4 remained only for approximately 15 min at the secondary ingression (Supplementary Fig. 3c). The localization of another Arp2/3 subunit (ARPC5) at

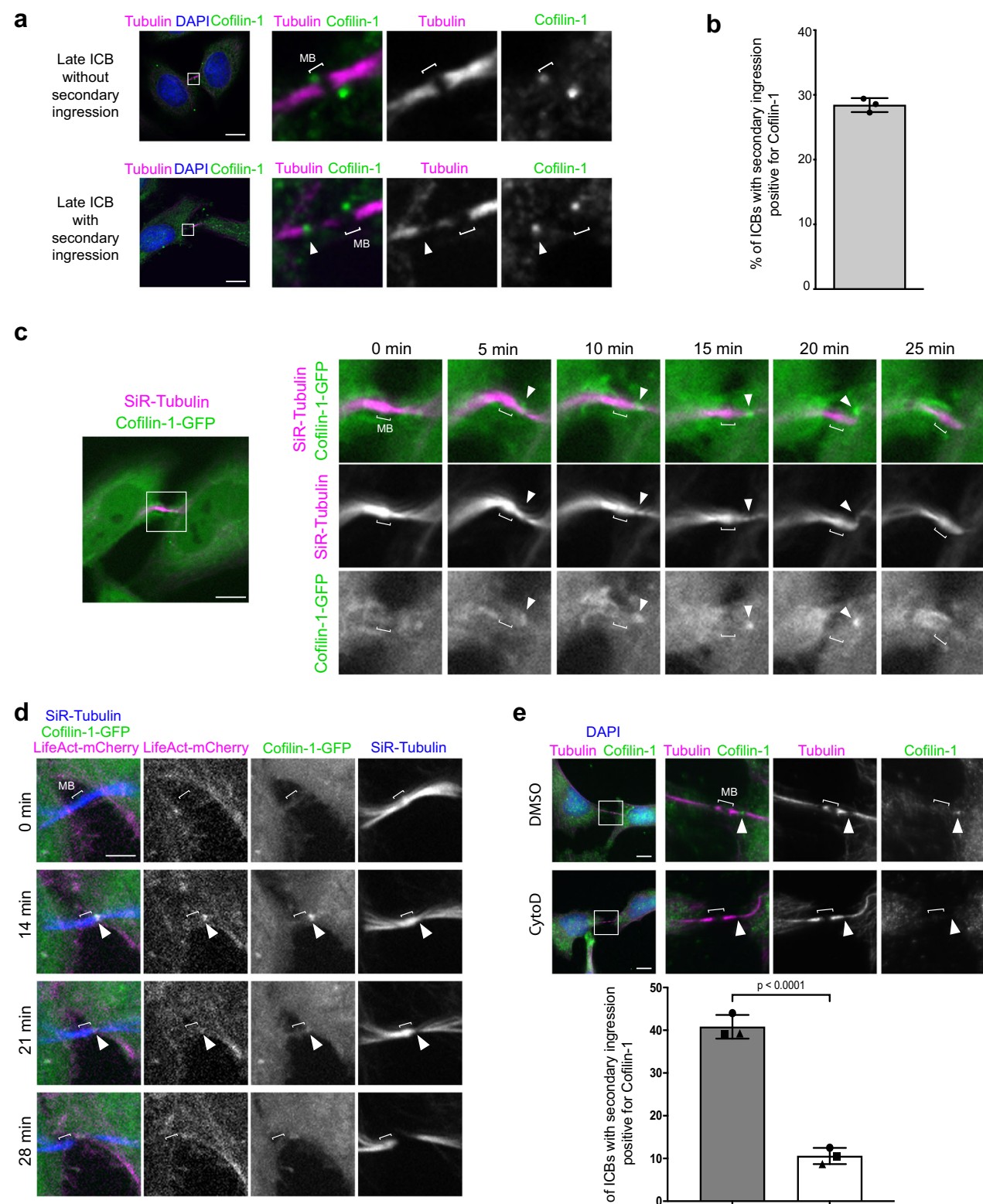

**Fig. 1 | Cofilin-1 and F-actin are transiently enriched at the secondary ingression. a** Staining of endogenous Cofilin-1 and α-tubulin + β-tubulin + acetylated tubulin (hereafter Tubulin) at indicated stages of cytokinesis in HeLa cells. DAPI: 4′,6-diamidino-2-phenylindole. Bracket (here and in all Figures): midbody (MB). Arrowhead: Cofilin-1 at the secondary ingression. Scale bars = 10 μm. **b** Percentage of ICBs with secondary ingression positive for Cofilin-1. Mean ± SD, *n* ≥ 15 cells per experiment, *N* = 3 independent experiments. **c** Snapshots of a spinning disk confocal microscopy movie of cells stably expressing Cofilin-1-GFP and incubated with fluorescent SiR-Tubulin. Arrowheads: Cofilin-1 at the secondary ingression. Scale

bar = 10 μm. **d** Snapshots of a spinning disk confocal microscopy movie of cells stably co-expressing Cofilin-1-GFP and LifeAct-mCherry and incubated with fluorescent SiR-Tubulin. Arrowheads: Cofilin-1 and LifeAct co-localization at the secondary ingression. Scale bar = 5 μm. **e** Top: Staining of endogenous Cofilin-1 and Tubulin in late ICBs with secondary ingression (arrowheads) in cells treated with either DMSO or 0.4 μM Cytochalasin D (CytoD) for 1 h prior to fixation. Scale bars = 10 μm. Bottom: Percentage of ICBs with secondary ingression positive for Cofilin-1. Mean ± SD, *n* ≥ 19 cells per condition, *N* = 3 independent experiments. Two-tailed unpaired Student's *t*-test.

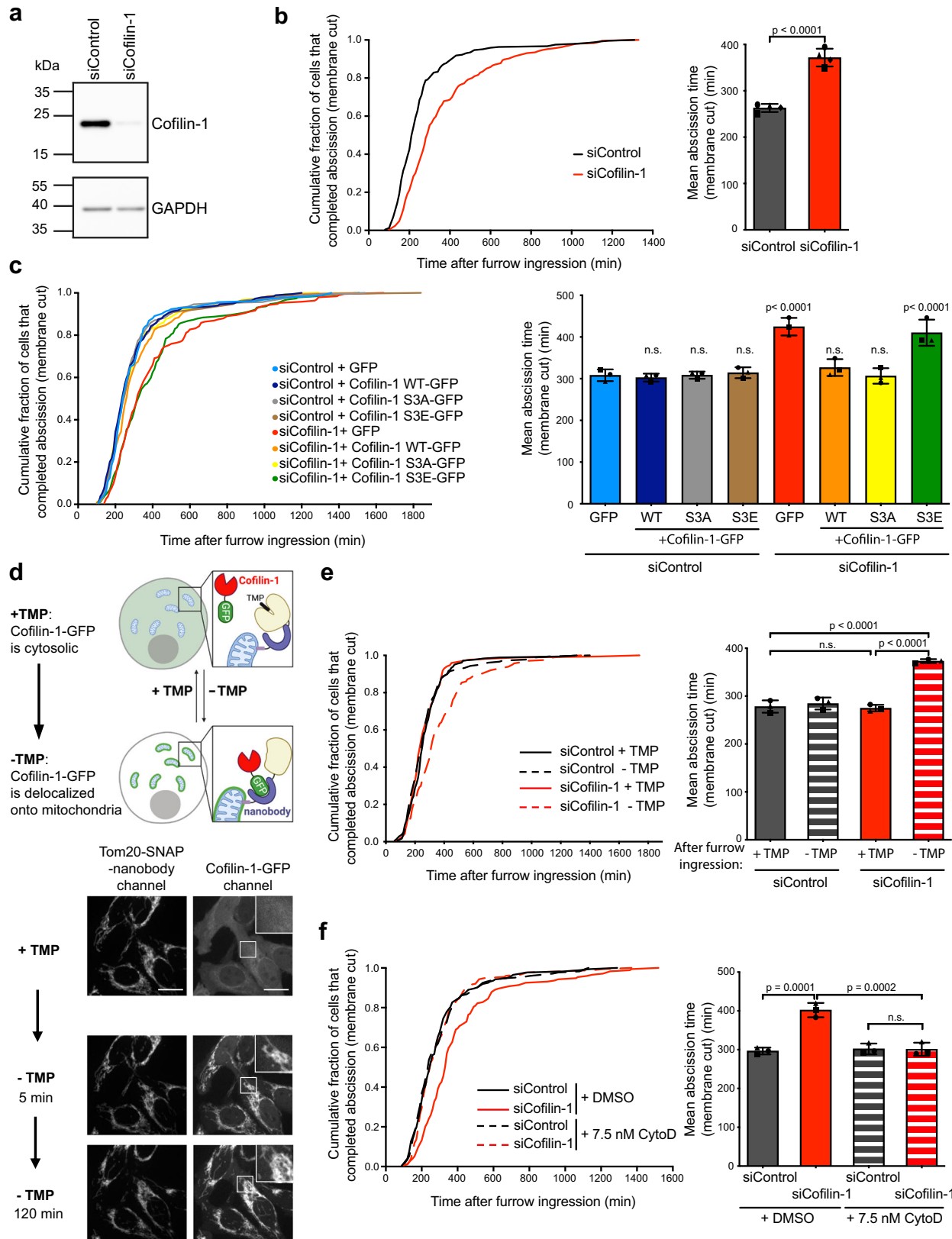

the secondary ingression and its disappearance at the time of the MT cut were confirmed in mouse 3T3 cells (Supplementary Fig. 3d). Of note, Arp2/3 can nucleate linear actin filaments upon activation by SPIN90[55]. However, we did not detect SPIN90 together with Arp2/3 or F-actin at the secondary ingression (Supplementary Fig. 3e). We conclude that the F-actin that transiently accumulates with Cofilin-1 at the

secondary ingression (Fig. 1d) likely corresponds to branched actin filaments that disassemble concomitantly with the MT severing.

We next investigated whether Cofilin-1 regulates the assembly/ disassembly of branched F-actin at the secondary ingression. Upon Cofilin-1 depletion, the endogenous Arp2/3 subunit ARPC2 was detected at the secondary ingression in a greater proportion of ICBs

**Fig. 2 | Cofilin-1 contributes to abscission by regulating F-actin depolymerization. a** Lysates of HeLa cells treated with either Control or Cofilin-1 siRNAs were blotted for endogenous Cofilin-1 and GAPDH (loading control). This experiment was repeated at least three times independently with similar results. **b** Left: Cumulative distribution of the fraction of cells that completed abscission (membrane cut measured by phase-contrast) as function of time after complete furrow ingression in cells treated with either Control or Cofilin-1 siRNAs ($n \geq 245$ cells per condition from 4 independent experiments). Abscission times (from complete furrow ingression to abscission) were determined using phase contrast time-lapse microscopy. Right: Mean abscission time (min) ± SD in indicated cells, $n \geq 59$ cells per condition, $N = 4$ independent experiments. Two-tailed unpaired Student's t-test. **c** Left: same analysis as in (**b**) for cells treated with either Control or Cofilin-1 siRNAs and transiently transfected with indicated constructs ($n \geq 183$ cells per condition from 3 independent experiments). Right: Mean abscission times (min) ± SD in indicated cells, $n \geq 60$ cells per condition, $N = 3$ independent experiments. One-way ANOVA with Tukey's multiple comparisons test. p-values after comparison to siControl + GFP alone are indicated. n.s. = non-significant ($p > 0.05$). **d** Top: Schemes illustrating how the removal of TMP induces the sequestration of Cofilin-1-GFP at the surface of the mitochondria upon activation of anti-GFP nanobodies. Adapted from ref. 49. Bottom: Snapshots of a spinning disk confocal microscopy movie of cells stably co-expressing Cofilin-1-GFP and the Tom20-anti-GFP nanobody-SNAP-DHFR fusion protein (labelled with SiR-SNAP) either when TMP is present or 5 and 120 min upon TMP removal, as indicated. Scale bars = 20 μm. **e** Left: same analysis as in (**b**) for cells stably co-expressing Cofilin-1-GFP and the Tom20-anti-GFP nanobody-SNAP-DHFR fusion protein, treated with either Control or Cofilin-1 siRNAs, upon removal (-) of TMP or not (+) after furrow ingression ($n \geq 164$ cells per condition from 3 independent experiments). Right: Mean abscission time (min) ± SD in indicated cells, $n \geq 38$ cells per condition, $N = 3$ independent experiments. One-way ANOVA with Tukey's multiple comparisons test. **f** Left: same analysis as in (**b**) for cells treated with either Control or Cofilin-1 siRNAs and incubated with either DMSO or 7.5 nM Cytochalasin D (CytoD) ($n \geq 173$ cells per condition from $N = 3$ independent experiments). Right: Mean abscission time (min) ± SD in indicated cells, $n \geq 51$ cells per condition, $N = 3$ independent experiments. One-way ANOVA with Tukey's multiple comparisons test. n.s. = non-significant ($p > 0.05$).

(Supplementary Fig. 3f), suggesting that ARPC2 remained longer at this location in the absence of Cofilin-1. Consistent with this hypothesis, live cell imaging in cells stably expressing GFP-ARPC4 showed an increase in the mean residence time of ARPC4 at the secondary ingression before the MT cut in Cofilin-1-depleted cells (Fig. 3d). Similar increase was observed in non-depleted cells treated with the F-actin stabilizing drug Jasplakinolide (Jasp) (Supplementary Fig. 3g), indicating that actin dynamics is important for Arp2/3 removal at the secondary ingression. Of note, Cofilin-1 depletion did not impact global F-actin levels in the ICB (Supplementary Fig. 3h), nor F-actin levels (Supplementary Fig. 3i) or Arp2/3 levels at the secondary ingression (Supplementary Fig. 3j).

Together, these results indicate the existence of a local pool of branched F-actin at the secondary ingression shortly before the MT cut and show that Cofilin-1 controls the disassembly of branched F-actin at this location.

## Cofilin-1 promotes abscission by favoring MT severing through ESCRT-III-mediated recruitment of spastin but does not impact the membrane cut occurring later

Since Cofilin-1, Arp2/3 and F-actin all appear transiently at the secondary ingression and Cofilin-1 depletion perturbs Arp2/3 dynamics before the MT cut, we hypothesized that Cofilin-1 may influence abscission by regulating MT severing. When we monitored the MT cut using fluorescent SiR-Tubulin and the membrane cut (actual abscission) using phase contrast microscopy in the same cells throughout cytokinesis, we noticed that these are two clearly separate events (Fig. 4a and Supplementary movie 3). MT severing at the secondary ingression and abscission are generally thought to occur simultaneously −or almost simultaneously− in cultured cells (see discussion). In contrast, we found that the MT cut happened on average 90 min after furrow ingression, whereas the membrane cut occurred on average 185 min later, with a broader time distribution in control cells (Fig. 4b siControl). Of note, for each individual cell, the MT cut always preceded the membrane cut but there was no correlation between the timing of these two events, suggesting that they are not directly coupled (Supplementary Fig. 4a). Silencing Cofilin-1 strongly delayed the MT cut compared to control cells (Fig. 4b, left panel solid curves and middle panel). A similar delay was observed upon chemical stabilization of actin filaments in non-depleted cells treated with Jasp (Supplementary Fig. 4b). Importantly, when we measured, for each cell, the time elapsed between the MT cut and the membrane cut, we did not observe any differences between the control- and Cofilin-1-depleted cells (Fig. 4b, right panel). These results indicate that the delay in abscission upon Cofilin-1 depletion is fully explained by a delay in cutting the MTs and suggest that Cofilin-1 does not play a role once the MTs are cut.

To demonstrate whether Cofilin-1 promotes abscission by controlling the MT cut, we used the drug Batabulin which was shown to efficiently depolymerize the stable MTs present in the ICB, except in the very central part of the midbody[13] (Fig. 4c). As previously reported[13], the addition of Batabulin after complete furrow ingression in control cells was compatible with cell progression up to final abscission (determined by phase contrast) with no delay compared to DMSO-treated cells (Fig. 4d). Importantly, the abscission delay observed upon Cofilin-1 depletion was fully rescued when MTs were depolymerized using Batabulin (Fig. 4d). We conclude that Cofilin-1 promotes abscission by favoring MT severing but is not involved in the subsequent steps leading to abscission.

We next investigated how Cofilin-1 influences the MT cut required for abscission. First, we observed that the transient Cofilin-1 pool present at the secondary ingression precisely localized at the tip of the ESCRT-III cone labeled by the CHMP4B subunit in fixed control cells (Fig. 5a). Since the tip of the cone corresponds to the place where spastin will sever the MTs, we then analyzed whether Cofilin-1 depletion might perturb ESCRT-III and spastin behavior. The kinetics of appearance of the CHMP4B cone was not altered upon Cofilin-1 depletion (Fig. 5b, Time T1) but it took significantly more time for the cones to eventually trigger the MT cut (Fig. 5b, Time T2). This was not due to cone instability or retraction, since i) the MT cut occurred at a normal distance from the midbody (Fig. 5c), ii) the CHMP4B cones grew regularly −albeit with a slower rate− (Fig. 5d). Similar reduction in the growth rate of CHMP4B cone was observed in non-depleted cells treated with Jasp (Supplementary Fig. 4c). Of note, we observed that the mean intensity of CHMP1B −the ESCRT-III component that recruits spastin during cytokinesis− was decreased in late ICBs with secondary ingression in Cofilin-1-depleted cells compared to control cell, but unchanged in earlier ICBs without secondary ingression (Fig. 5e). As a consequence, the downstream recruitment of spastin was reduced in late ICBs with secondary ingression but was unaffected in earlier ICBs (Fig. 5f). Consistent with Cofilin-1 functioning upstream to spastin, the depletion of spastin did not impact on Cofilin-1 recruitment at the tip of the ESCRT-III cone (Supplementary Fig. 4d). We thus conclude that Cofilin-1 regulates the rate of ESCRT-III cone elongation and promotes abscission by controlling local MT severing through CHMP1B-dependent spastin recruitment at the secondary ingression.

## Branched F-actin promotes abscission by preventing excessive ESCRT-III cone elongation and favoring MT severing

We next wondered whether the presence of the branched F-actin pool at the secondary ingression was required for abscission. In normal fixed dividing cells, the Arp2/3 subunit ARPC2 localized precisely at the tip of the ESCRT-III cone labeled by its subunit IST1 (Fig. 6a). As

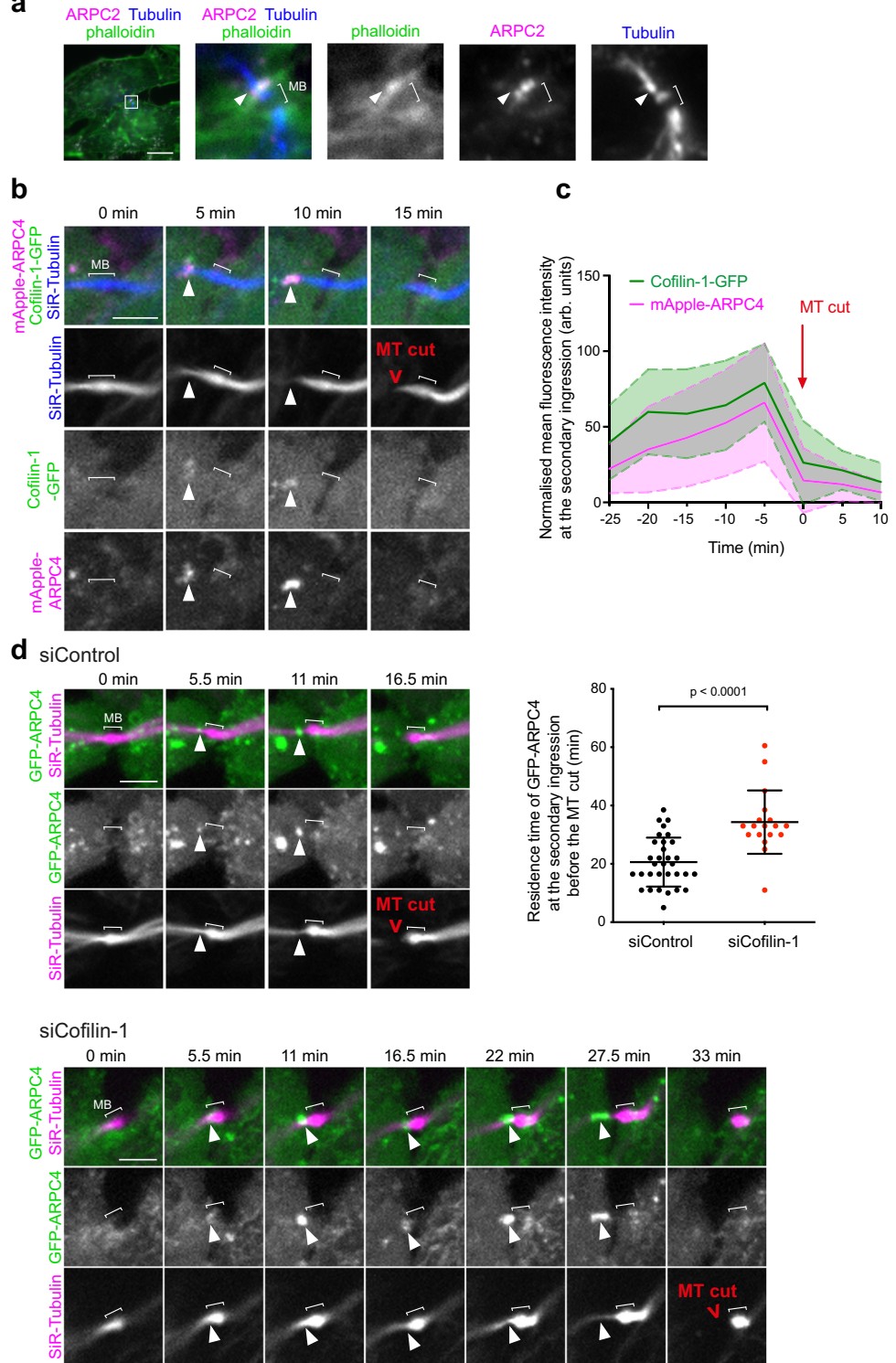

**Fig. 3 | Cofilin-1 colocalizes with a pool of branched F-actin at the secondary ingression before the MT cut and regulates the lifetime of this pool. a** Staining of endogenous ARPC2, Tubulin and F-actin (labelled by fluorescent phalloidin). Scale bar = 10 µm. Arrowheads: ARPC2 and F-actin co-localization at the secondary ingression. This experiment was repeated at least three times independently with similar results. **b** Snapshots of a spinning disk confocal microscopy movie of cells stably co-expressing Cofilin-1-GFP and mApple-ARPC4 and incubated with fluorescent SiR-Tubulin. Arrowheads: Cofilin-1 and ARPC4 co-localization at the secondary ingression. Scale bar = 5 µm. **c** Intensity profile of Cofilin-1-GFP (green) and mApple-ARPC4 (magenta) at the secondary ingression as a function of time in cells described in (**b**). Intensity values (arbitrary units) were normalized to the maximum

intensity observed for each cell analyzed. Mean ± SD, *n* = 12 cells. All movies were registered with respect to the time of the MT cut (t0, red arrow). **d** Top left and bottom: Snapshots of a spinning disk confocal microscopy movie of cells stably expressing GFP-ARPC4 and incubated with fluorescent SiR-Tubulin after treatment with Control (top) or Cofilin-1 (bottom) siRNAs. Arrowheads: ARPC4 localization at the secondary ingression. Scale bar = 5 µm. Top right: The time elapsed between ARPC4 appearance at the secondary ingression and the MT cut was measured in the aforementioned conditions. Mean ± SD in indicated cells, *n* = 32 (siControl) or 18 (siCofilin-1) cells per condition from *N* = 4 independent experiments. Two-sided Mann-Whitney test.

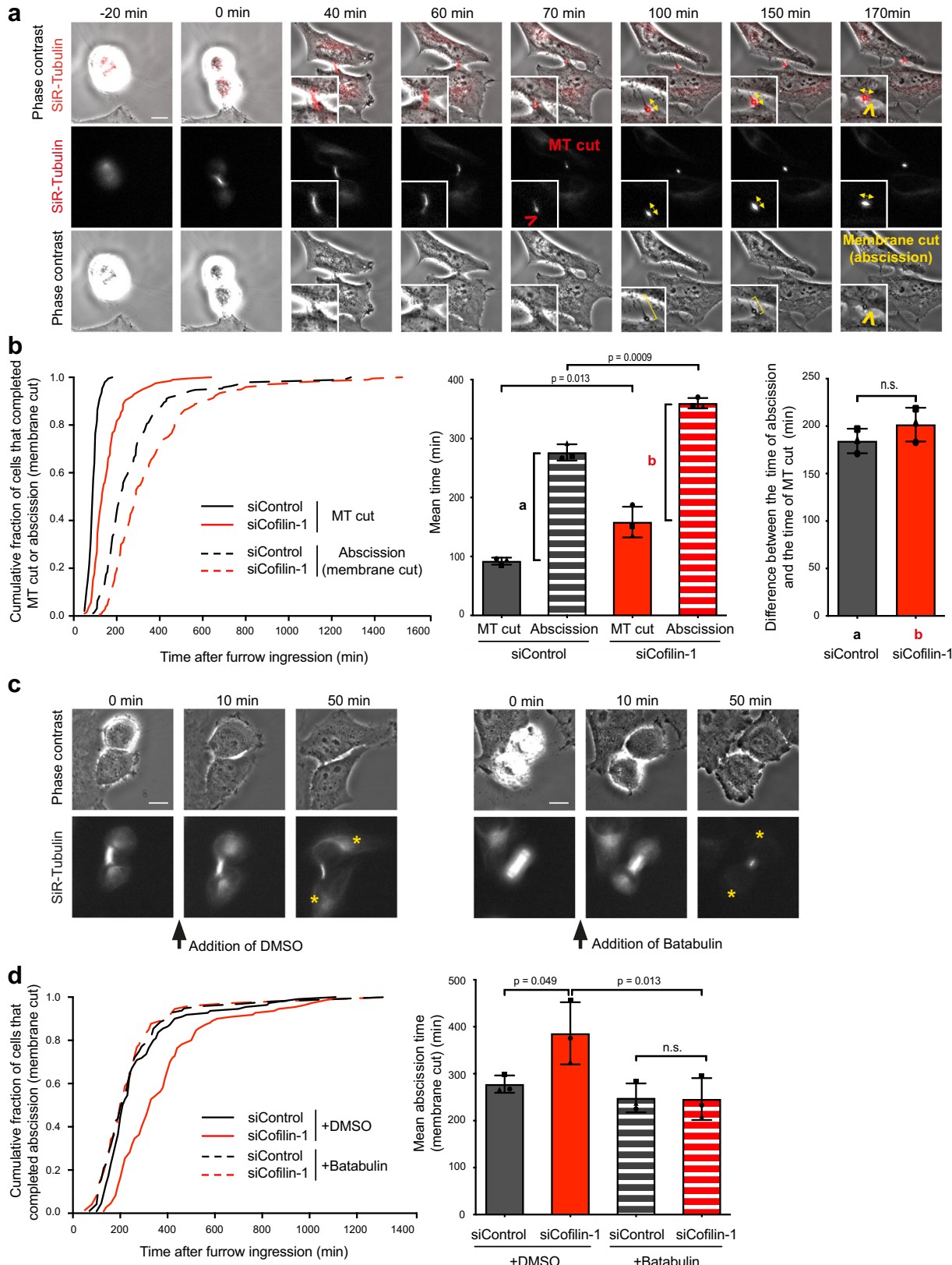

expected, ARPC4 and F-actin transiently accumulated at the tip of the ESCRT-III cone a few minutes before the MT cut (Fig. 6b and Supplementary Fig. 5a, arrowheads). Functionally, preventing the formation of branched F-actin by adding the Arp2/3 inhibitor CK666 after furrow ingression resulted in delayed abscission compared to DMSO-treated cells (Fig. 6c). Furthermore, Arp2/3 inhibition delayed the MT cut after

ESCRT-III cone formation but left unchanged the time elapsed between the MT cut and the membrane cut (Fig. 6d and Supplementary Fig. 5b), indicating that once the MTs are cut, abscission occurs normally after Arp2/3 inhibition. Similar results were obtained in cells treated with high doses of either CytoD or LatA (Supplementary Fig. 5c, d). Importantly, the abscission delay due to Arp2/3 inhibition could be

**Fig. 4 | Cofilin-1 promotes abscission by favoring MT severing but does not influence the membrane cut occurring later. a** Snapshots of a widefield microscopy movie of HeLa cells incubated with fluorescent SiR-Tubulin and recorded both in phase contrast and in fluorescence. Time 0 corresponds to the time of complete furrow ingression. Red arrowhead: Microtubule cut ($t = 70$ min). Yellow brackets indicate the plasma membrane of the un-severed ICB. Yellow arrowhead: Membrane cut = physical abscission ($t = 170$ min). Yellow double-headed arrows: orientation of the midbody visualized by SiR-Tubulin. Note the rotation of the midbody at the time of the membrane cut. Scale bar = 10 μm. **b** Left: Cumulative distribution of the fraction of cells that completed MT cut (based on the SiR-tubulin channel, solid curves) and abscission (based on the phase contrast channel, dashed curves) as function of time after complete furrow ingression, in cells treated with either Control or Cofilin-1 siRNAs ($n \geq 190$ cells per condition from 3 independent experiments). MT cut and abscission timings were determined in the same cells using the fluorescent SiR-Tubulin channel and the phase contrast time-lapse channel, respectively. Middle: Mean MT cut and abscission time (min) ± SD in indicated cells, $n \geq 61$ cells per condition, $N = 3$ independent experiments. Two-tailed unpaired Student's t-test. a and b represent the time difference between the abscission and the MT cut in Control and Cofilin-1-depleted cells, respectively. Right: a and b values defined in the middle panel have been represented. Mean a and b time difference (min) ± SD in indicated cells, $n \geq 61$ cells per condition, $N = 3$ independent experiments. Two-tailed unpaired Student's t-test. n.s. = non-significant ($p > 0.05$). **c** Snapshots of a widefield microscopy movie of HeLa cells incubated with fluorescent SiR-Tubulin and recorded both in phase contrast and in fluorescence, after addition of either DMSO (left) or 2 μM Batabulin (right) after furrow ingression (between $t = 0$ and $t = 10$ min). Yellow stars: SiR-Tubulin labelling in the cell bodies after treatment. Note the complete disappearance of MTs (except in the central part of the midbody) upon Batabulin treatment. Scale bars = 10 μm. **d** Left: Cumulative distribution of the fraction of cells that completed abscission (based on the phase contrast channel) as function of time after complete furrow ingression, in cells treated with either Control or Cofilin-1 siRNAs and incubated or not with Batabulin after complete furrow ingression, as indicated ($n \geq 73$ cells per condition from 3 independent experiments). Right: Mean abscission time (min) ± SD in indicated cells, $n \geq 16$ cells per condition, $N = 3$ independent experiments. One-way ANOVA with Tukey's multiple comparisons test. n.s. = non-significant ($p > 0.05$).

fully rescued by depolymerizing the MTs with Batabulin (Fig. 6e), showing that branched F-actin specifically favors the MT cut during abscission.

To get further mechanistic insights, we next investigated whether Arp2/3 inhibition changed the behavior of spastin at the secondary ingression. We recorded spastin dynamics during cytokinesis by taking advantage of an inducible cell line that expressed low levels of GFP-spastin[56]. In control cells, spastin first accumulated at the midbody, then localized as a cone on one side of the midbody —consistent with spastin being recruited by CHMP1B there— and eventually concentrated at the tip of the cone before the MT cut (Fig. 7a and Supplementary movie 4). Arp2/3 inhibition led to a delay between the arrival of spastin at the cone and the MT cut (Fig. 7a and Supplementary movie 4), in agreement with the observed delay in the MT cut (Fig. 6d) and with a greater proportion of ICBs with endogenous spastin detected at the secondary ingression upon Arp2/3 inhibition (Supplementary Fig. 6a). Consistent with Arp2/3 acting upstream to spastin, the depletion of spastin did not prevent ARPC2 recruitment at the tip of the ESCRT-III cone (Supplementary Fig. 6b). Furthermore, the mean intensity of endogenous spastin (as well as CHMP1B) at the secondary ingression was not modified but spastin localization appeared more elongated (Fig. 7b, top and bottom left panels and Supplementary Fig. 6c). We also noticed that spastin abnormally accumulated at the constriction site far away from the midbody in CK666-treated cells as compared to DMSO-treated cells (Fig. 7b, top and bottom right panels). Altogether, we conclude that the delay in the MT cut observed upon Arp2/3 inhibition is associated with an abnormal spastin distribution (shape and distance from the midbody) at the secondary ingression.

Since spastin is recruited by CHMP1B and concentrates at the tip of the ESCRT-III cone, these results led us to hypothesize that Arp2/3 may control the behavior of ESCRTs at the secondary ingression. Consistently, abnormally long CHMP1B-positive cones were observed upon Arp2/3 inhibition (Fig. 7c). These long CHMP1B-positive cones were also observed upon actin depolymerization using high doses of either CytoD or LatA (Supplementary Fig. 6d, e), arguing that branched F-actin normally sets the dimension of the ESCRT-III cone that polymerize from the midbody to the secondary ingression. To further investigate the role of Arp2/3 in restraining the extension of the ESCRT-III cone, we simultaneously recorded in live cells Arp2/3 (with mApple-ARPC4), ESCRT-III (with CHMP4B-GFP) and MTs (with SiR-Tubulin) as a function of time upon CK666 addition. In randomly picked control (DMSO-treated) cytokinetic cells that displayed a pool of Arp2/3 at the tip of the ESCRT-III cone at the beginning of the movie (time 0), we observed a limited growth of the cone before MTs were severed (Supplementary movie 5, Fig. 7d and Fig. 7e-i each black curve

corresponds to an individual cell). MT severing occurred within approximately 15 min (Fig. 7e-ii), at a distance of approximately 2 μm from the midbody (Fig. 7e-iii). In contrast, in cells treated with CK666, we observed 1) a disappearance of the Arp2/3 pool at the tip of the cone and 2) an abnormal growth of the ESCRT-III cone due to ESCRT-III polymerization for an extended period of time before MT were eventually severed (Supplementary movie 5, Fig. 7f and Fig. 7e-i green curves). MT severing occurred typically 30-70 min after the addition of the drug (Fig. 7e-ii), at a distance of approximately 4 μm (Fig. 7e-iii). In addition, ESCRT-III cones detached more frequently from the midbody in CK666-treated cells (26.9 %, $n = 52$) compared to DMSO-treated cells (9.8 %, $n = 41$) (Chi$^2$ contingency test: $p = 0.037$). We propose that the Arp2/3-positive branched F-actin pool present at the tip of the ESCRT-III cone limits its extension, thereby preventing the sliding of spastin along the ICB and favors productive MT severing close to the midbody. This likely explains why in the absence of Arp2/3, MTs are not severed properly and thus abscission is delayed.

## Discussion

We reveal here a role of Arp2/3-dependent branched F-actin together with Cofilin-1 in promoting cytokinetic abscission through the specific regulation of MT severing at the secondary ingression.

To characterize the function of Cofilin-1 and Arp2/3 in abscission, we monitored the MT cut and the membrane cut in the same dividing cells and realized that these two events are far from being simultaneous or almost simultaneous, as generally assumed in cultured cells. Reported measurements of abscission timing give various mean values depending on the technique that is used, even for a single cell type such as HeLa cells: (1) time-lapse imaging of fluorescently labelled tubulin[18,27,32,57–62]; (2) measurement of cytoplasmic exchange of photoactivable proteins between the two daughter cells, which ceases after abscission or at least when the ICB diameter is small enough[13,63]; or (3) time-lapse visualization of the physical cut of the ICB (membrane cut) using phase contrast microscopy[19,28–30,64,65]. The observation that the MT cut and the membrane cut are two separated events likely explains the discrepancy in the timing of abscission between studies reported in the literature.

Whether F-actin has a function in the final steps of cytokinesis, after furrow ingression, is a long-standing question. The fact that F-actin must be globally disassembled within the ICB to promote abscission is well established[19,28,29,32,33]. In contrast, an active role of F-actin in abscission, likely locally and/or transiently, is still debated. In a seminal study describing the constricting role of ESCRT-III at the secondary ingression, the addition of the F-actin polymerization inhibitor Latrunculin B (LatB) after furrow ingression did not alter the timing of abscission[13]. In contrast, adding the F-actin polymerization

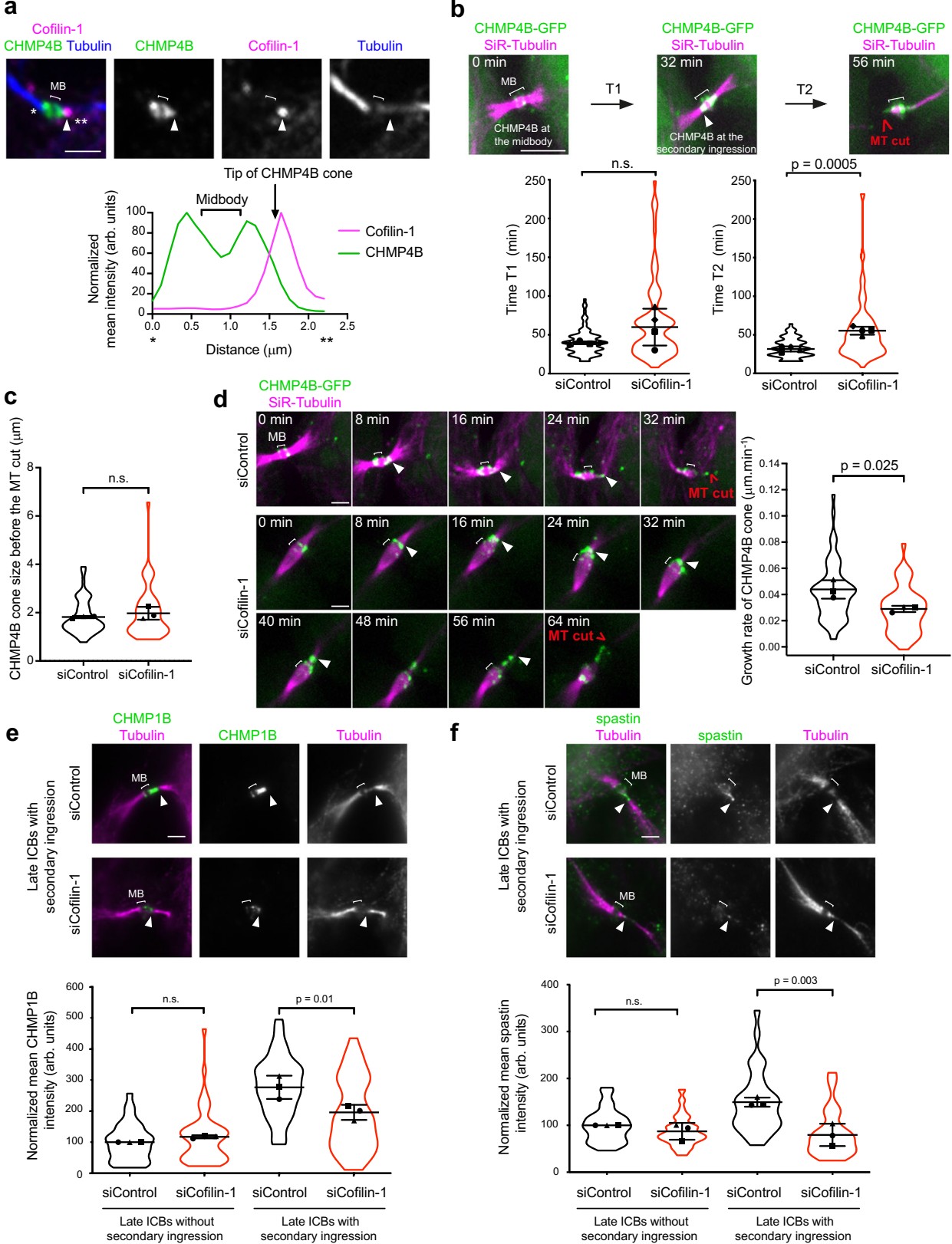

inhibitor Cytochalasin D after furrow ingression increased binucleated cells formation due to bridge regression after midbody formation, indicating that F-actin is required for late ICB stability and/or completion of abscission[34]. Furthermore, incubation with Blebbistatin after furrow ingression prevented the formation of the secondary ingression, indicating a role of myosin II and, presumably, F-actin in this

process[66]. Failure to see a role of F-actin in late cytokinetic steps in some studies likely stems from differences in drugs or doses that have been used or exact time of addition after furrow ingression. In particular, LatB is less efficient than LatA and is likely inactivated by serum[67], which might explain why defects on abscission were not previously detected[13]. Here, we found similar effects of relatively high doses of

**Fig. 5 | The MT severing delay observed in Cofilin-1-depleted cells is associated with a defective recruitment of the ESCRT-III subunit CHMP1B and spastin.**
**a** Top: Staining of endogenous Cofilin-1, CHMP4B and Tubulin in a late ICB with secondary ingression (arrowhead). Scale bar = 2 μm. Bottom: Intensity profile of CHMP4B (green) and Cofilin-1 (magenta) between * and ** along the ICB from the top merge picture. This experiment was repeated at least three times independently with similar results. **b** Top: Snapshots of a spinning disk confocal microscopy movie of cells stably expressing CHMP4B-GFP and incubated with fluorescent SiR-Tubulin. T1: time from CHMP4B-GFP recruitment at the midbody ($t = 0$ min) to appearance at the secondary ingression (arrowhead); T2: time from CHMP4B-GFP appearance at the secondary ingression to the MT cut. Scale bar = 5 μm. Bottom: Mean T1 and T2 (min) ± SD in indicated cells, $n = 10-35$ cells per condition, $N = 4$ independent experiments (violin plots). Two-tailed unpaired Student's t-test with Welch's correction. n.s. = non-significant ($p > 0.05$). **c** Quantification of CHMP4B-GFP cone size at the timepoint just before the cut of the MTs from the movies described in (**d**). Mean ± SD, $n = 8-32$ cells per condition, $N = 3$ independent experiments (violin plots). Two-tailed unpaired Student's t-test. n.s. = non-significant ($p > 0.05$). **d** Left: Snapshots of a spinning disk confocal microscopy movie (8 min intervals) of cells stably expressing CHMP4B-GFP and incubated with fluorescent SiR-Tubulin and treated with either Control or Cofilin-1 siRNAs. Arrowhead: tip of the ESCRT-III cone. Scale bar = 2 μm. Right: Quantification of the growth rate of CHMP4B cone in indicated cells. Mean ± SD, $n = 8-32$ cells per condition, $N = 3$ independent experiments (violin plots). Two-tailed unpaired Student's t-test. **e** Top: Representative images of endogenous CHMP1B and Tubulin in cells treated with either Control or Cofilin-1 siRNAs. Arrowhead: secondary ingression. Scale bars = 5 μm. Bottom: mean fluorescence intensity of CHMP1B (arbitrary units) at the midbody (late ICBs without secondary ingression) or at the midbody + secondary ingression (late ICBs with secondary ingression) in indicated cells. Mean ± SD, $n = 7-21$ cells per condition, $N = 3$ independent experiments (violin plots). Intensities are normalized in each experiment to the mean intensity of siControl cells in late ICB without secondary ingression. One-way ANOVA with Tukey's multiple comparisons test. n.s. = non-significant ($p > 0.05$). **f** Same as in (**e**) for endogenous spastin. Mean ± SD in indicated cells, n = 8–20 cells per condition, $N = 3$ independent experiments. One-way ANOVA with Tukey's multiple comparisons test. n.s. = non-significant ($p > 0.05$).

CytoD and LatA on abscission, MT cut and ESCRT-III polymerization. Of note, continuous incubation of dividing cells with CK666 did not result in binucleated cells, suggesting that Arp2/3 is not required for furrow ingression nor for ICB stability, but whether it perturbed abscission had not been investigated[33]. Here, we used time-lapse phase contrast microscopy to visualize the actual physical cut of the ICB and treated the cells with CK666 after furrow ingression. This approach demonstrated that branched F-actin promotes abscission.

Based on our localization data and functional analysis, we further involved two actin partners, the Arp2/3 complex and the actin disassembling protein Cofilin-1, in cytokinetic abscission. Using live cell imaging, we discovered a previously overlooked, transient pool of Cofilin-1 and Arp2/3, which appear together at the secondary ingression 15–20 min before severing of the MTs. Furthermore, the disappearance of Cofilin-1 and Arp2/3 at the secondary ingression occurs simultaneously with the MT cut —according to our time-resolution of 5 min— suggesting that these two events are tightly coupled. A spike of F-actin at the secondary ingression in late cytokinetic steps has been previously reported[34,66]. However, the nature of this actin pool, its relationship with MT disassembly and its precise role in abscission remained unknown. We speculate that this pool likely corresponds to the Arp2/3-F-actin pool that we characterized here. Further work would be needed to investigate whether actin and tubulin crosslinkers known to localize at the secondary ingression, such as the protein Gas2l3[68,69], might regulate the assembly of the branched actin pool at this location and thereby promote the MT cut. Importantly, this actin pool precisely assembles at the tip of ESCRT-III cones. Of note, two components of Arp2/3 (ARPC3 and ARPC5L) and Cofilin-1 have been recently identified by mass spectrometry upon coimmunoprecipitation with CHMP4B during cytokinesis[70]. These interactions, either direct or indirect, might account for the observed overlap of Arp2/3 and Cofilin-1 with the ESCRT-III machinery at the secondary ingression.

How Arp2/3 and F-actin transiently accumulate at the secondary ingression? Our data argue that Arp2/3 is not activated by SPIN90 — which promotes Arp2/3-dependent linear F-actin[55]— suggesting that Arp2/3 nucleates branched F-actin at the secondary ingression. Future investigations will nevertheless be required to determine how Arp2/3 is activated to assemble branched F-actin at this site. Since Cofilin-1 is continuously present with Arp2/3 and F-actin at the secondary ingression for 15–20 min before the MT cut, it is likely that a balance of Arp2/3 activators and Cofilin-1 regulates branched F-actin turnover at this location. Consistent with the known function of Cofilin-1 in controlling F-actin disassembly and dissociation of Arp2/3-mediated actin branches[47,71–75], we found that Cofilin-1 depletion increases the residence time of Arp2/3 at the secondary ingression. This is in agreement with the discrete localization of Cofilin-1 at the secondary ingression.

Importantly, low doses of the F-actin depolymerization drugs Latrunculin-A or Cytochalasin D (which have no effect on abscission in control cells) rescue the abscission delay observed in Cofilin-1-depleted cells, indicating a function of Cofilin-1 in remodeling F-actin for proper abscission. Of note, Cofilin-1 does not increase F-actin levels all over the ICB. This contrasts with a global regulation of Arp2/3 and F-actin in the whole ICB by the oxidoreductase MICAL1[29,76], which is known to synergize with Cofilin-1 to disassemble F-actin in vitro[77–79]. Thus, MICAL1 and Cofilin-1 appear to play distinct role during cytokinetic abscission. Of note, Arp2/3 acutely disappears from the secondary ingression concomitantly with the MT cut. This could be explained by an increase in Cofilin-1 activity or by the inactivation of Arp2/3 activators at the time of the MT severing.

Mechanistically, we propose that Arp2/3 together with Cofilin-1 controls the timing of abscission by favoring the severing of the MTs at the secondary ingression (Fig. 8). In Cofilin-1-depleted cells and in CK666-treated cells, the recruitment and the behavior of spastin is impaired, respectively. Upon Cofilin-1 depletion, the residence time of Arp2/3 is increased, and the growth rate of ESCRT-III cones is decreased. This is associated with a reduction of CHMP1B at the ICB, which likely explains the observed defective recruitment of spastin and the delay in the MT cut. Interestingly, stabilizing F-actin using Jasplakinolide reduces the growth rate of ESCRT-III cones, delays the MT cut and increases the residence time of Arp2/3, as observed upon Cofilin-1 depletion. Cofilin-1 depletion might thus reduce the dynamics of the branched F-actin pool at the tip of the ESCRT-III cone. We speculate that in Cofilin-1-depleted cells, the presence of a non-dynamic pool of F-actin at this location creates local resistance that slows down the elongation of the ESCRT-III cone and prevents normal CHMP1B incorporation. In contrast, removing Arp2/3 by adding CK666 does not impair spastin recruitment but profoundly alters its behavior, which also results in delaying MT severing. Using live and fixed cell imaging, we found that spastin is first recruited at the midbody, then accumulates at the tip of the ESCRT-III cone where it promotes local MT severing. This reveals spastin dynamics during cytokinesis. Either CK666, Cytochalasin D or Latrunculin A treatments have a strong effect on the geometry of the ESCRT-III cone, which grows continuously from the midbody over several micrometers for an extended period of time. As a consequence, spastin at the tip of the cone continuously progresses away from the midbody. We hypothesize that this does not give enough time for spastin to sufficiently depolymerize MT at a given location, which thus delays MT severing. Reinforcing the notion of an abnormal spastin-mediated severing in CK666-treated cells, we noticed that bridge MTs appear to fade over time upon CK666 treatment, suggesting continuous depolymerization, as previously observed in spastin-depleted cells[27]. At the mesoscopic level, we

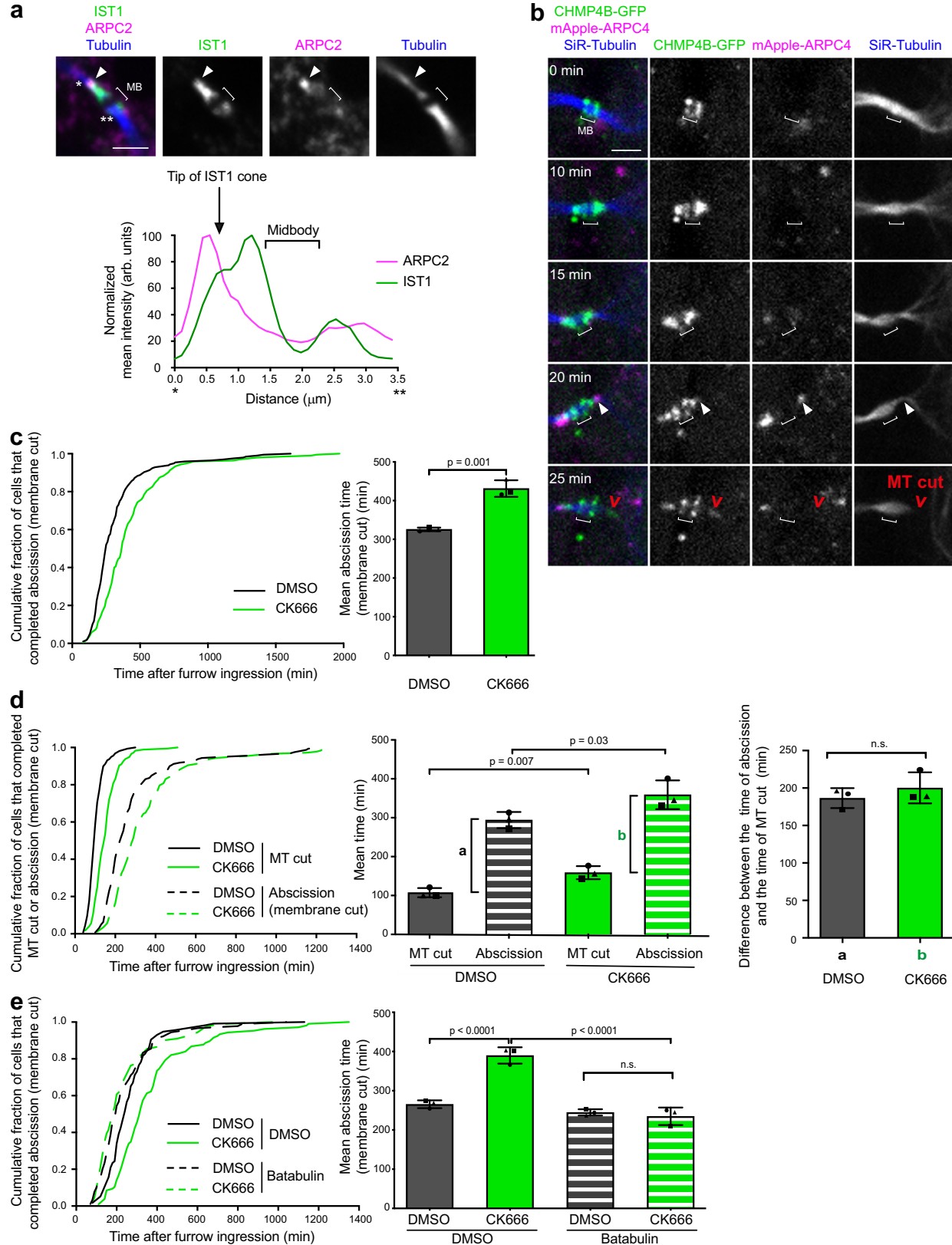

speculate that the branched F-actin meshwork acts as a physical barrier that blocks the abnormal extension of the ESCRT-III polymers. To our knowledge, the Arp2/3 pool at the tip of the ESCRT-III cone is the first reported regulatory component that sets the dimensions of the ESCRT-III cone, which appears critical to control spastin activity. Altogether, these results highlight the local requirement of Cofilin-1

and Arp2/3-dependent branched F-actin in abscission by controlling the correct recruitment and stability of the spastin pool at the secondary ingression.

In summary, our results demonstrate a role of the actin cytoskeleton in the late steps of cytokinesis. Indeed, we found that branched F-actin and Cofilin-1 were specifically required for the MT cut. This

**Fig. 6 | Branched F-actin promotes abscission by favoring MT severing. a** Top: Staining of endogenous ARPC2, IST1 and Tubulin in a late ICB with secondary ingression (arrowhead). Scale bar = 2 μm. Bottom: Intensity profile of ARPC2 (magenta) and IST1 (green) between * and ** along the ICB from the top merge picture. This experiment was repeated at least three times independently with similar results. **b** Snapshots of a spinning disk confocal microscopy movie of cells stably co-expressing CHMP4B-GFP and mApple-ARPC4 and incubated with fluorescent SiR-Tubulin. Arrowheads: ARPC4 localization at the tip of the CHMP4B cone. Scale bar = 2 μm. **c** Left: Cumulative distribution of the fraction of cells that completed abscission (determined by phase contrast time-lapse microscopy) as function of time after complete furrow ingression in cells treated, after furrow ingression, with either DMSO or CK666 ($n \geq 195$ cells per condition from 3 independent experiments). Note that DMSO and 160 μM CK666 were added after furrow ingression in synchronized cells (see Methods). Right: Mean abscission time (min) ± SD in indicated cells, $n \geq 61$ cells per condition, $N = 3$ independent experiments. Two-tailed unpaired Student's $t$-test. **d** Same analysis as in Fig. 4b in cells treated with either DMSO or CK666 after complete furrow ingression, as indicated. Left: $n \geq 150$ cells per condition from 3 independent experiments. Middle and right: $n \geq 41$ cells per condition, $N = 3$ independent experiments. Two-tailed paired Student's t-test. n.s. = non-significant ($p > 0.05$). **e** Same analysis as in Fig. 4d in cells treated with either DMSO or CK666, together with DMSO or 2 μM Batabulin after complete furrow ingression, as indicated. Left: $n \geq 89$ cells per condition from 3 independent experiments. Right: $n \geq 21$ cells per condition, $N = 3$ independent experiments. One-way ANOVA with Tukey's multiple comparisons test. n.s. = non-significant ($p > 0.05$).

reveals a crosstalk between the actin and MT cytoskeleton via ESCRT polymers. We conclude that cytokinetic abscission can be divided into two distinct steps, a MT cut and a membrane cut, which are controlled by distinct molecular machineries.

## Methods

### Cell cultures
HeLa CCL-2 (ATCC) and NIH3T3 cells were grown in Dulbecco's Modified Eagle Medium (DMEM) GlutaMax (31966; Gibco, Invitrogen Life Technologies) supplemented with 10% fetal bovine serum and 1× penicillin−streptomycin (Gibco) in 5% $CO_2$ at 37 °C. BMEL cells[80] were grown in RPMI 1640 GlutaMax medium (61870010; Gibco, Invitrogen Life Technologies) supplemented with 10% fetal bovine serum, 1× penicillin−streptomycin (Gibco), 50 ng.ml$^{-1}$ epidermal growth factor, 30 ng.ml$^{-1}$ insulin-like growth factor II (PeproTech), 10 ng.ml$^{-1}$ insulin (Roche) and cultured on Collagen I (BD Biosciences) coated dishes with 5 % $CO_2$ at 37 °C. CHMP4B-LAP-GFP; Cofilin-1-LAP-GFP; Cofilin-1-LAP-GFP + LifeAct-mCherry; Cofilin-1-LAP-GFP + mApple-ARPC4; CHMP4B-LAP-GFP + mApple-ARPC4 and CHMP4B-LAP-GFP + LifeAct-mCherry stable cell lines were generated by lentiviral transduction of HeLa ATCC CCL-2 cells and selected by FACS. Cofilin-1-LAP-GFP + Tom20-anti-GFP nanobody-SNAP-DHFR$_{F98}$ stable cell lines were generated by electroporating HeLa ATCC cells stably expressing Cofilin-1-LAP-GFP (see above) with the corresponding plasmid described in ref. 49, followed by G418 selection (Gibco) and FACS. These cells were continuously grown with 10 μM TMP (Trimethoprim, Sigma Aldrich). HeLa cells that stably express doxycycline-inducible EGFP-spastin wild-type M87 were a kind gift from T. Kapoor and were described in ref. 56. EGFP-spastin expression was induced using 1 μg.ml$^{-1}$ doxycycline for 20–30 h before imaging. NIH3T3 cells that stably express GFP-ARPC5 were a kind gift from Alexis Gautreau. For Cytochalasin D (Sigma-Aldrich) treatment, cells were incubated with 0.4 μM of drugs for 60 min before fixation.

### Transfections, siRNAs and plasmid constructs
pEGFP-N1 human cofilin WT / S3A / S3E were a kind gift from James Bamburg (Addgene plasmid # 50859 / # 50860 / # 50861). Cofilin-1, CHMP4B, LifeAct and ARPC4 CDS cDNAs were subcloned into Gateway pENTR plasmids and introduced into transient expression vectors or lentiviral constructs by LR recombination (Thermo Fisher) to generate EGFP, mCherry or mApple fusion proteins. siRNA-resistant versions of Cofilin-1 have been obtained by mutating 6 bp of the siRNA-targeting sequence using QuickChange (Agilent). Plasmids were transiently transfected in HeLa cells for 48 h using X-tremeGENE 9 DNA reagent (Roche).

For silencing experiments, HeLa cells were transfected with 25 nM siRNAs (siCofilin-1) or 50 nM siRNAs (siSpastin) for 72 h using HiPerFect (Qiagen) or Lipofectamine RNAiMAX (Invitrogen), following the manufacturer's instructions. siRNAs against Luciferase (used as control, 5′CGUACGCGGAAUACUUCGA[dU][dU]3′), Cofilin-1 (5′C CUCUAUGAUGCAACCUAU[dU][dU]3′)[81] and spastin (5′AAACGG ACGUCUAUAAUGA[dU][dU]3′)[82] have been synthetized by Sigma. In rescue experiments, cells were first transfected for 72 h with siRNAs using HiPerFect, then transfected by plasmids encoding GFP or siRNA-resistant Cofilin-1-GFP wild type or mutant using DharmaFECT Duo transfection reagent (Dharmacon) for an additional 24 h.

### Western blots
Cells were lysed on ice for 30 min in NP-40 lysis buffer (50 mM Tris, pH 8, 150 mM NaCl, 1% NP-40 and protease inhibitors) and centrifuged at 10,000 g for 10 min. Migration of 20 μg of lysate supernatants was performed in 4−12% gradient or in 10% SDS-PAGE gels (Bio-Rad Laboratories), transferred onto PVDF membranes (Millipore), blocked and incubated with corresponding antibodies in 5% milk in 50 mM Tris-HCl pH 8.0, 150 mM NaCl, 0.1% Tween20, followed by horseradish peroxidase-coupled secondary antibodies (1:10,000, Jackson ImmunoResearch) and revealed by chemiluminescence (GE Healthcare).

### Immunofluorescence and image acquisition
HeLa cells were grown on coverslips and fixed either with paraformaldehyde (PFA) 4% for 20 min at room temperature or with ice-cold methanol for 2−3 min (see Supplementary Table 1). Cells were then permeabilized with 0.1% Triton-X100 when necessary, blocked with PBS containing 1% bovine serum albumin (BSA) and successively incubated for 1 h at room temperature with primary (see Supplementary Table 1) and secondary antibodies (1:500, Jackson Laboratory) diluted in PBS containing 1% BSA before DAPI staining for 5 min (0.5 mg/ml, Serva). Cells were mounted in Fluoromount G (Southern Biotech). For endogenous staining of ARPC2, cells were first fixed with 4% PFA + 0.5% Triton-X100 for 1.5 min before incubation with 4% PFA for 15 min.

Images were acquired with an inverted TiE Nikon microscope, using a x60 or a ×100 1.4 NA PL-APO objective lens and MetaMorph software (MDS) driving either a CCD camera (Photometrics Coolsnap HQ) or a Retiga R6 CCD camera (Teledyne Photometrics). Alternatively, images were acquired on an inverted Nikon Eclipse Ti-E microscope equipped with a CSU-X1 spinning disk confocal scanning unit (Yokogawa) coupled to either a Prime 95 S scientific complementary metal-oxide semiconductor (sCMOS) camera (Teledyne Photometrics) or a EMCCD Camera (Evolve 512 Delta, Photometrics). Images were then converted into RGB 8-bit images using ImageJ software (NIH) and resampled to 300ppi using Adobe Photoshop (bicubic method).

### Image and data analysis
All images and movies were analyzed with the Fiji software. Mean intensity values were extracted from manually drawn ROIs for each channel of interest using Fiji and mean background intensity was subtracted. The scan lines were obtained using an ROI (straight line), with a width corresponding to the bridge after subtraction of the background intensity. Intensity values were normalized to control condition or to the maximal intensity measured as indicated in the

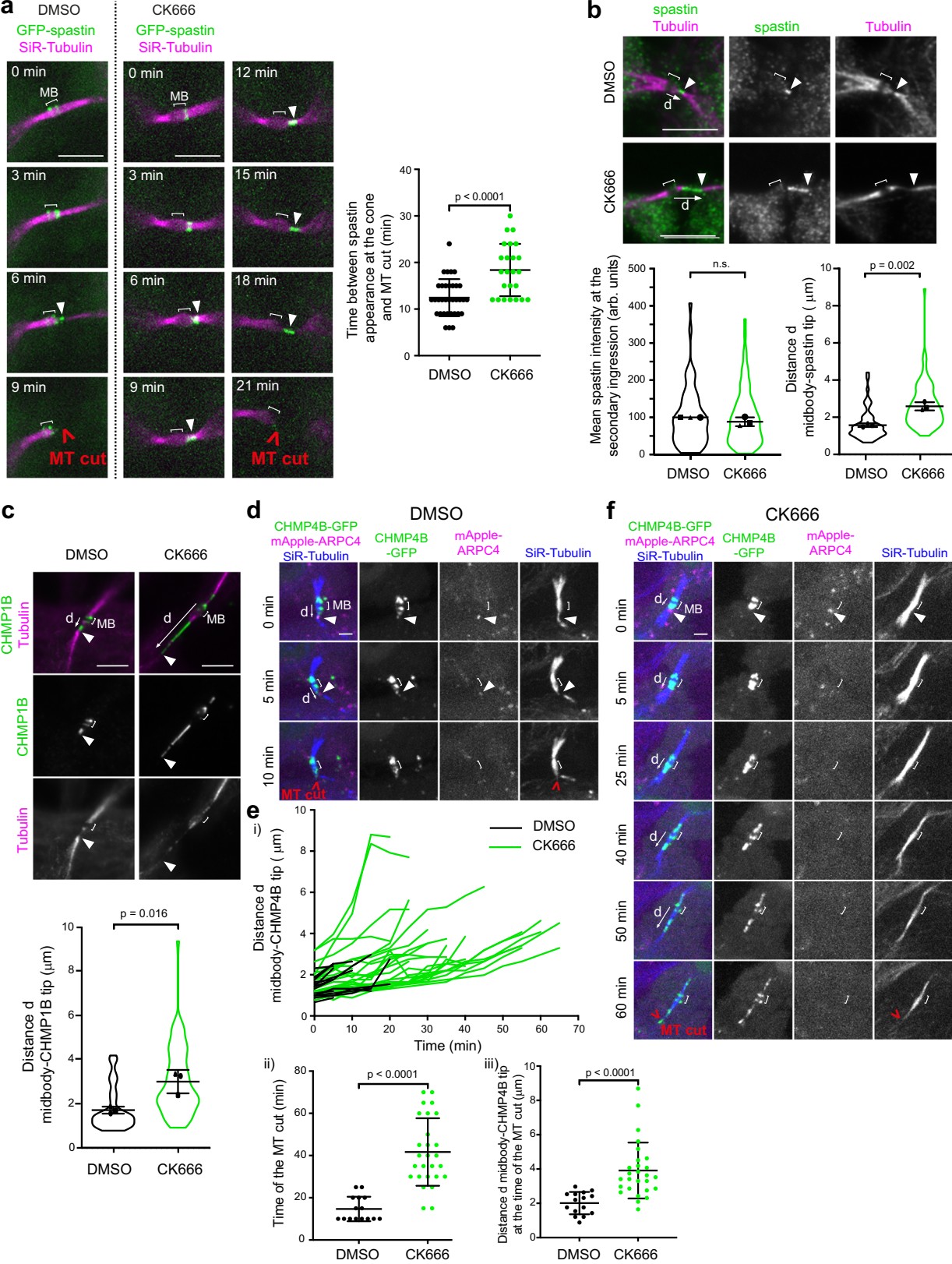

figure legends. To measure the rate of CHMP4B cone growth, the size of the CHMP4B-GFP cone −from the midbody ring to the tip of the cone− was measured every 5–8 min using CHMP4B-GFP movies, from the appearance of the cone to the occurrence of the MT cut. The slope of the curve was used to determine the mean rate of ESCRT-III cone elongation.

Illustrations were created with BioRender.com.

### Time-lapse microscopy

For time-lapse phase-contrast imaging, cells were plated for 48 h on glass bottom 12-well plates (MatTek) and put in an open chamber (Life Imaging) equilibrated in 5% $CO_2$ and maintained at 37 °C. Cells were

**Fig. 7 | Branched F-actin prevents excessive ESCRT-III cone elongation and promotes correct spastin localization. a** Left: Snapshots of a spinning disk confocal microscopy movie of cells stably expressing inducible GFP-spastin M87 and incubated with fluorescent SiR-Tubulin, either in the presence of DMSO or 200 μM CK666. Arrowheads: spastin at secondary ingressions. Scale bars= 5 μm. Right: Time (min) elapsed between spastin appearance at the cone and the MT cut. Mean ± SD in indicated cells, $n$ = 30 (DMSO) or 24 (CK666) cells per condition from $N$ = 4 independent experiments. Two-sided Mann-Whitney test. **b** Top: Staining of endogenous spastin and Tubulin in late ICBs with secondary ingression (arrowhead) after addition of either DMSO or 200 μM CK666. Scale bars = 5 μm. Bottom left: Mean intensity (arbitrary units) of spastin at the secondary ingression. The intensities are normalized in each experiment to the mean intensity of DMSO-treated cells. Mean ± SD, $n$ = 12–43 cells per condition, $N$ = 3 independent experiments (violin plots). Two-tailed unpaired Student's t-test with Welch's correction. n.s. = non-significant ($p$ > 0.05). Bottom right: distance d (μm) between the center of the midbody and the tip of the spastin labelling. Mean ± SD, $n$ = 9–38 cells per condition, $N$ = 3 independent experiments (violin plots). Two-tailed unpaired Student's t-test. **c** Top: Staining of endogenous CHMP1B and Tubulin in late ICBs with secondary ingression after addition of either DMSO or 200 μM CK666. Arrowheads: tip of the CHMP1B cone. Scale bars = 5 μm. Bottom: distance d (μm) between

the center of the midbody and the tip of the CHMP1B cone. Mean ± SD, $n$ = 12–23 cells per condition, $N$ = 3 independent experiments (violin plots). Two-tailed unpaired Student's t-test. **d** Snapshots of a spinning disk confocal microscopy movie of cells stably co-expressing CHMP4B-GFP and mApple-ARPC4, incubated with fluorescent SiR-Tubulin and treated with DMSO added between $t$ = 0 and $t$ = 5 min. Arrowheads: ARPC4 localization at the tip of the CHMP4B cone. Scale bar = 2 μm. **e** (i) Distance d (μm) between the center of the midbody and the tip of the CHMP4B-GFP cone as a function of time from cells recorded as in (**d**) and treated with either DMSO (black curves) or 200 μM CK666 (green curves). Each line represents a single cell and only cells displaying a dot of ARPC4 at the tip of the CHMP4B cone at the beginning of the movies ($t$ = 0 min) were depicted. Drugs were added between t = 0 and t = 5 min. $n$ = 15 (DMSO) or 27 (CK666) cells. The last point of each curve corresponds to the time of the MT cut. (ii) Time (min) between the start of the movies described in (i) and the MT cut. Mean ± SD, $n$ = 15 (DMSO) or 27 (CK666) cells. Two-sided Mann-Whitney test. (iii) Distance d between the center of the midbody and the tip of the CHMP4B-GFP cone at the timepoint preceding the MT cut from movies described in (i). Mean ± SD, n = 15 (DMSO) or 27 (CK666) cells. Two-sided Mann-Whitney test. **f** Same as in (**d**) but after treatment with 200 μM CK666. Scale bar = 2 μm.

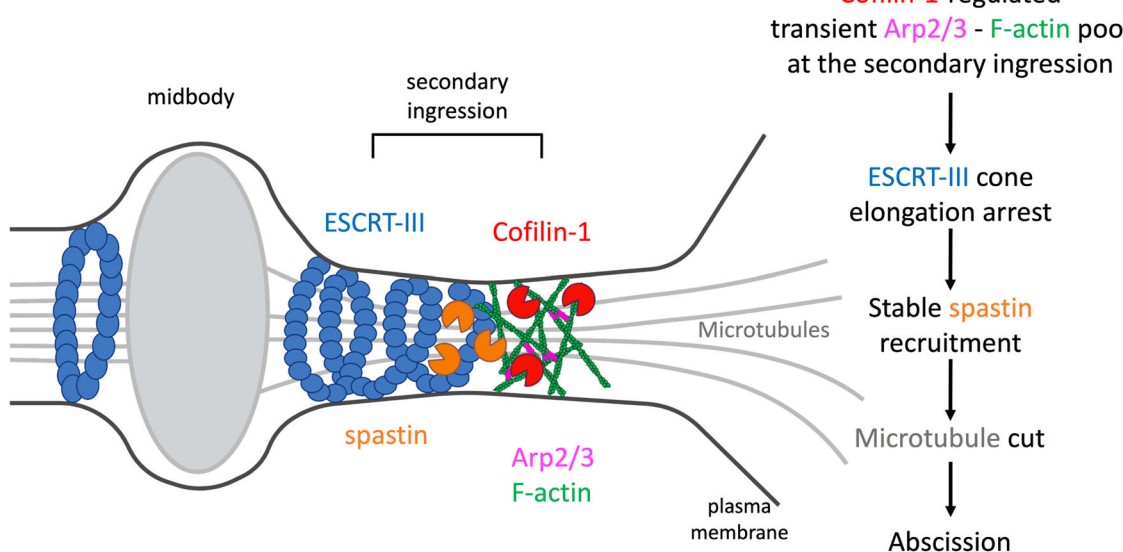

**Fig. 8 | Model: control of cytokinetic abscission by Cofilin-1, Arp2/3 and branched F-actin-dependent MT severing.** Both Cofilin-1 and Arp2/3 promote abscission by regulating the cut of the ICB MTs at the secondary ingression. Cofilin-1 controls the growth rate of the ESCRT-III cone and the disassembly of a transient pool of Arp2/3-mediated branched F-actin at the tip of the ESCRT-III cone at the secondary ingression at the time of the MT cut. In the absence of Cofilin-1, this pool

stays longer, the ESCRT-III cone elongation is slowed down and CHMP1B-dependent recruitment of spastin is reduced. In the absence of Arp2/3, the ESCRT-III cone continuously extends away from the midbody which impairs stable spastin recruitment and leads to abscission defects due to delayed MT severing. We propose that Arp2/3 limits the extension of the ESCRT-III cone by acting as a physical barrier at the secondary ingression, thus favoring the MT cut.

recorded every 10 min for 48 h using an inverted Nikon Eclipse TiE microscope with a ×20 0.45 NA Plan Fluor ELWD or a x40 0.95 NA PL-APO objectives controlled by Metamorph software (Universal Imaging).

To determine abscission times, we used the phase contrast channel and measured the time elapsed from the first image after complete furrow ingression until the occurrence of the membrane cut of the ICB, which is usually accompanied by a sudden change in orientation of the midbody. To determine the MT cut time, the culture medium was replaced and 25–100 nM of fluorescent SiR-tubulin (SC002, Spirochrome) were added at least 6 h before recording. We then measured both the MT cut and the membrane cut in the same cells, starting from complete furrow ingression up to the occurrence of the first cut of the microtubules on one side of the midbody visualized by the SiR-tubulin signal in the fluorescent channel.

Drugs were added (2 μM Batabulin or 160 μM CK666) or washed out (10 μM TMP = Trimethoprim) after complete furrow ingression using two protocols: In Figs. 4d, 6d, e, S5c and S5d mitotic cells were randomly picked and recorded for 24 h every 10 min and drugs were added between timepoint 30 and 40 min, thus after completion of furrow ingression; In Figs. 2e and 6c, cells were synchronized by mitotic shake: from 80% confluent flasks, a first mitotic shake was performed to remove debris and dead cells. One hour later, a second mitotic shake was performed to harvest mitotic cells. Metaphase cells were then plated on a collagen-coated glass surface and let to adhere at 37 °C. Next, the medium with or without drugs was changed and cells were recorded 60 min after replating every 10 min for 24 h on randomly picked cells. Only cells with already fully ingressed furrow were analyzed (Figs. 2e and 6c).

To visualize protein localization by time-lapse fluorescent microscopy, cells were recorded with an inverted Nikon Eclipse Ti-E

microscope equipped with a CSU-X1 spinning disk confocal scanning unit (Yokogawa) and with either an EMCCD (Evolve 512 Delta, Photometrics) or a Prime 95 S sCMOS camera (Teledyne Photometrics). Images were acquired with 100 × 1.4 NA PL-APO or 60 × 1.4 NA PL-APO VC objectives and MetaMorph software (MDS). 25–100 nM of fluorescent SiR-Tubulin were added and incubated at least 6 h before recording. SNAP-Cell 647-SiR substrate (S9102S New England Biolabs) was used to reveal SNAP-tag following the manufacturer's instructions. For live cell experiment presented Fig. 7a, d, f DMSO or 200 µM CK666 was added between the first and the second timepoint after starting the acquisition.

## Statistical analyses

All plots and statistical tests were performed using the GraphPad Prism software. The presented values are displayed as mean ± SD from at least three independent experiments and the test used is indicated in the figure legends. In all statistical tests, $p$-value > 0.05 was considered as nonsignificant. $P$-values are indicated in the figures.

## Reporting summary

Further information on research design is available in the Nature Portfolio Reporting Summary linked to this article.

## Data availability

All data are available in the main text or the supplementary materials. Uncropped gels are presented in the Source Data file and in Supplementary Fig. 7. All reagents described in this paper will be made available to readers and be sent upon request. Source data are provided with this paper.

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

## Acknowledgements

We thank G. Romet-Lemonne, J. Mathieu, Antoine Jegou and the Echard Lab members N. Gupta-Rossi and M. Serres for critical reading of the manuscript; N. Gupta-Rossi and N. Sassoon for BMEL cells; Quentin Gaia Gianetto for Supplementary Fig. 1a; A. Gautreau and T. Kapoor for reagents; K. Johnsson for kindly sharing reagents before publications; and the Recombinant antibodies platform (TAb-IP, Institut Curie, Paris) for antibodies. We thank PH. Commere from the Utechs CB, Institut Pasteur for FACS sorting. This work has been supported by Institut Pasteur, CNRS, ANR (Cytosign, RedoxActin) and Fondation pour le Recherche Médicale: Recherche soutenue par la FRM EQU202103012627 to A.E. T.A. received a post-doctoral fellowship from Fondation pour le Recherche Médicale (FRM SPF201809006907).

## Author contributions

T.A. conceived, carried out and analyzed the experiments presented in Figs. 1, 2, 3, 4, 5, 6, 7 and Supplementary Fig. 1, 2, 3, 4, 5, 6; S.F. obtained preliminary results on Cofilin-1 localization; A.E. conceived and supervised the project; A.E. secured funding; T.A. and A.E. wrote the manuscript.

## Competing interests

The authors declare no competing interests.
