## [Peer Review File · Nature Communications]

REVIEWER COMMENTS

Reviewer #1 (Remarks to the Author):

This is an interesting, and well executed study that adds new mechanistic insight to the mechanisms governing cytokinetic abscission of human tissue culture cells. Using high resolution live cell microscopy, Advedissian and colleagues show that F-actin, Arp2/3 complex components and the F-actin disassembly factor, Cofilin-1, transiently localize (for 20 min) at the presumptive abscission site known as the “secondary ingression” site. Intriguingly, depletion of Cofilin-1 delayed abscission, in a manner that could be restored using low dose Latrunculin A, and also extended the residence time of ARPC4 (by 10-15 min). Through a series of elegant experiments, the abscission delay upon cofilin- depletion was convincingly attributed to a delay in MT severing (known to be dependent on spastin), and NOT on the subsequent membrane scission events that occur later. Indeed, differences in the timing of, and the requirements for, MT severing versus membrane cutting (i.e. true abscission), lead the authors to propose these are two quite distinct and sequential steps in the process. This is an important conclusion of the work, that, as the authors point out, may help to reconcile reported differences in “abscission” timing between studies, depending on how “abscission” was scored. It is clear from this work that MT severing should not be used as a read-out for “abscission”, which actually can occur much later.

Given the importance of the process being studied, the high quality of the data and the resulting mechanistic advances made, I recommend publication in Nature Communications. However, I do have the following questions, comments, and suggestions for further improving the manuscript prior to publication:

1) Why was CytoD used to reduce F-actin for the set of experiments in Fig 1 (GFP-Arp2 localization) while LatA was used in the experiments of Fig 2 (abscission timing)? These two drugs have different mechanisms of action (F-actin depolymerization, versus G-actin sequestration). This seems important, especially given what is stated in the Discussion: “In a seminal study describing the constricting role of ESCRT-III at the secondary ingression, the addition of the F-actin polymerization inhibitor Latrunculin B after furrow ingression did not alter the timing of abscission (ref. 13). In contrast, adding the F-actin polymerization inhibitor Cytochalasin D after furrow ingression increased binucleated cells formation due to bridge regression after midbody formation, indicating that F-actin is required for late ICB stability and/or completion of abscission (ref. 34).... Failure to see a role of F-actin in late cytokinetic steps in some studies likely stems from differences in drugs or doses that have been used.” Given these statements and the apparent central importance of this newly-identified pool of F-actin at the secondary ingression site, it would seem appropriate to directly compare the two drugs in all of the assays (localization and abscission timing), rather than using a different drug in different assays. If there are differences between CytoD and LatA, this could be mechanistically informative and should be declared.

2) Low dose LatA is shown to restore abscission timing in cofilin-1-depleted cells, while the authors suggest “that the branched F-actin meshwork acts as a physical barrier that blocks the abnormal extension of the ESCRT-III polymers”. So, what about higher doses of LatA (of course added after

midbody formation)? If the model is true, one would expect delayed MT-severing and delayed abscission, whether cofilin-1 is present or not.

3) On the subject of F-actin drugs, I cannot help but wonder what the effects of low doses of the F-actin stabilising drug, jasplakinolide (or SiR-actin, which is essentially jasplakinolide), are on MT-severing, abscission timing, GFP-ARPC4 localization etc.. in control and cofilin-1-depleted cells. I do not feel this is a requirement for publication, but it could certainly provide further mechanistic insight and strengthen the model. Have the authors looked at this at all?

4) siCofilin-1 extended the residence time of GFP-ARPC4, but did this result in enhanced GFP-ARPC4 accumulation above WT levels? This looks to be the case from the examples shown in Fig. 3d, but is this true at the population level? Or was there a fixed quantity of ARPC4 recruited, which simply persisted for longer?

5) Fig. 2 After TMP removal (5 min or 120 min), sequestration of Cofilin-1-GFP at the mitochondria is clearly shown, and this is an exciting and elegant approach. However, it would strengthen the data to show and quantify the presumed depletion at the intercellular bridge, which is the relevant pool for the accompanying data on abscission timing. Also, in Fig. 2e: when exactly was TMP removed? It would be helpful to label it on the graph and specify in the legend.

6) The abstract lacks any methodology. I understand space is tight, but at the very least the model system should be mentioned.

7) Also in the Abstract: "Mechanistically, branched F-actin functions as a physical barrier that limits ESCRT-III cone elongation and thereby favors stable spastin recruitment." This sentence is too speculative for the abstract and should be restricted to the Discussion.

Typos

p4 "non-phosphorylable Cofilin-1" should be non-phosphorylatable Cofilin-1

p8 "significatively" should be "significantly"

Reviewer #2 (Remarks to the Author):

The manuscript submitted by Advedissian, Fremont and Echard highlight a role for the actin binding proteins Cofilin-1 and ARPC4 in the mechanism of cytokinetic abscission. This is a well written manuscript that provides several new important insights to the process of abscission. First, they show that microtubule severing and membrane cutting are temporally separated, as previously suggested from in vivo studies. Second, they reveal a role for cofilin-1 in late cytokinesis and demonstrate that depolymerization of actin at late abscission stages can rescue the abscission delay observed in cell depleted of Cofilin-1. Last, and perhaps most striking are their findings that the Arp2/3 protein ARPC4 has a role in positioning the ESCRT-III filament at the site of microtubules and membrane scission suggesting that microtubule, actin and ESCRT act in concert to determine the site of abscission. Overall, I find this work significant and suitable for publication in Nature communications. I have several comments that should be addressed prior to publication.

Specific comments:

1. The data presented in Fig. 1 indicate that cofilin levels peaks ~15 minutes prior to MT cut. Yet, in figure 3c the peak is ~5 minutes prior to MT cut. Can the authors please explain these differences?
2. I am having a hard time consolidating the cofilin-1 and ARPC4 findings. While I find the data presented for ARPC4 and the model that branched actin determine the site of ESCRT-III positioning, spastin recruitment and ultimately MT severing and abscission very convincing it is not clear to me how cofilin fit to this scenario. I guess the data support an upstream effect for cofilin. However, depolymerization of the actin pool at the constriction site by cofilin would suggest a downstream effect. This point should be clarified either by performing additional experiments to reveal the relationships between cofilin and ARPC4 at the intercellular bridge or by providing a reasonable explanation in the discussion that is well aligned with the data presented and with the known function of these proteins.
3. The authors show that cofilin and ARPC4 affect spastin recruitment. Based on these findings they conclude that these proteins function upstream to spastin. To substantiate this conclusion, the arrival and organization of these actin binding proteins to the intercellular bridge should be examined in cells depleted of spastin.
4. Few of the intercellular bridges shown in the figures has abnormal morphologies or atypical CHMP4B organization. Fig 3d, microtubule staining looks abnormal. Fig 6b 20 min, CHMP4B staining looks abnormal, Fig 7 bridge microtubules in cells treated with CK666 appear to fade over time suggesting continuous depolymerization. If these images reflect abnormalities resulting from the manipulations the cells were subjected to, it should be clearly stated in the text. Otherwise, these images should be replaced with more representative examples.

5. It was previously shown that the actin/tubulin cross linking protein Gas2l3 localize to the construction site of the intercellular bridge (Pe'er et al. PLOS One, 2013). It will be really interesting to examine whether this protein plays a role in the ARPC4 dependent spastin and ESCRT-III localization shown here. At the minimum this paper should be referred to in the discussion chapter.

Minor comments:

1. Because abscission timing was measured based on phase contrast and/or microtubule labeling it is important to specify on each plot what was measured to avoid confusion. This is relevant to all plots in Fig 2, Fig 4d, Fig 6 c, e.

2. Titles in Figures 3, 4, 5 should be toned down to reflect the data presented in each figure and avoid over interpretation.

3. The abstract statement: "contrary to the common assumption" is a bit misleading. It is correct that the field generally referred to microtubule severing as an indicative for abscission, but the notion that membrane severing itself occurs later is not new to the field and has been previously discussed. Please remove this statement from the abstract or tone it down considerably.

Response to Reviewers

" Cytokinetic abscission requires actin-dependent microtubule severing "

We thank both Reviewers for their very positive and constructive evaluation of our work. We believe that the added experimental data and Reviewer's suggestions helped us to further support the conclusions of the manuscript.

We addressed all comments raised by the Reviewers, most often with new experimental data, and provide below a full response.

New results have been added as **new main Fig. 1c, 2f and 5d** and **new Supplementary Fig. S1g, S2d, S2e, S3b, S3g, S3i, S3j, S4b, S4c, S4d, S5c, S5d, S6b and S6e**. Changes in the text have been highlighted.

Reviewer #1

This is an interesting, and well executed study that adds new mechanistic insight to the mechanisms governing cytokinetic abscission of human tissue culture cells. Using high resolution live cell microscopy, Advedissian and colleagues show that F-actin, Arp2/3 complex components and the F-actin disassembly factor, Cofilin-1, transiently localize (for 20 min) at the presumptive abscission site known as the "secondary ingression" site. Intriguingly, depletion of Cofilin-1 delayed abscission, in a manner that could be restored using low dose Latrunculin A, and also extended the residence time of ARPC4 (by 10-15 min). Through a series of elegant experiments, the abscission delay upon cofilin- depletion was convincingly attributed to a delay in MT severing (known to be dependent on spastin), and NOT on the subsequent membrane scission events that occur later. Indeed, differences in the timing of, and the requirements for, MT severing versus membrane cutting (i.e. true abscission), lead the authors to propose these are two quite distinct and sequential steps in the process. This is an important conclusion of the work, that, as the authors point out, may help to reconcile reported differences in "abscission" timing between studies, depending on how "abscission" was scored. It is clear from this work that MT severing should not be used as a read-out for "abscission", which actually can occur much later. Given the importance of the process being studied, the high quality of the data and the resulting mechanistic advances made, I recommend publication in Nature Communications. However, I do have the following questions, comments, and suggestions for further improving the manuscript prior to publication:

1) Why was CytoD used to reduce F-actin for the set of experiments in Fig 1 (GFP-Arp2 localization) while LatA was used in the experiments of Fig 2 (abscission timing)? These two drugs have different mechanisms of action (F-actin depolymerization, versus G-actin sequestration). This seems important, especially given what is stated in the Discussion: "In a seminal study describing the constricting role of ESCRT-III at the secondary ingression, the addition of the F-actin polymerization inhibitor Latrunculin B after furrow ingression did not alter the timing of abscission (ref. 13). In contrast, adding the F-actin polymerization inhibitor Cytochalasin D after furrow ingression increased binucleated cells formation due to

bridge regression after midbody formation, indicating that F-actin is required for late ICB stability and/or completion of abscission (ref. 34)... Failure to see a role of F-actin in late cytokinetic steps in some studies likely stems from differences in drugs or doses that have been used.” Given these statements and the apparent central importance of this newly-identified pool of F-actin at the secondary ingression site, it would seem appropriate to directly compare the two drugs in all of the assays (localization and abscission timing), rather than using a different drug in different assays. If there are differences between CytoD and LatA, this could be mechanistically informative and should be declared.

In the first version of the manuscript, we used high doses of CytoD (0.4 μ M) for strong and short term (1 h) depolymerization of F-actin to show that Cofilin-1 recruitment at the secondary ingression depends on F-actin. In contrast, we used low doses of LatA (20 nM) for long term (24 h movies) and partial reduction of F-actin to restore normal abscission timing in Cofilin-1-depleted cells. This choice was only based on historical reasons: we already knew which range of concentrations of LatA were low enough not to affect abscission in control cells. We indeed previously used this trick to restore normal F-actin levels and abscission timing when F-actin abnormally accumulates in bridges upon either OCRL1 depletion (Dambournet et al. *Nature Cell Biology* 2012) or MICAL1 depletion (Frémont et al. *Nature Communications* 2017).

To directly address the Reviewer's comment, we determined which concentration of CytoD (7.5 nM) did not affect abscission timing in control cells but was sufficient to fully rescue the abscission delay observed upon Cofilin-1 depletion (**new Fig. 2f**). Conversely, we increased LatA concentrations (100 nM) to acutely depolymerize F-actin in ICB and found that it reduces Cofilin-1 recruitment at the secondary ingression (**new Fig. S1g**), as previously observed with CytoD (Fig. 1e). In addition, LatA treatment led to an overgrowth of the ESCRT-III cone (**new Fig. S6e**), as previously found with CytoD (Fig. S6d).

We noticed that LatA at 100 nM —note that too much LatA leads to cell rounding and cannot be used for our analyzes— is less efficient than CytoD at 0.4 μ M to depolymerize F-actin in bridges (see representative picture in **new Fig. S1g**). This correlates with slightly less potent effects of LatA on Cofilin-1 recruitment and ESCRT-III cone overgrowth, as compared to CytoD.

In summary, the conclusions based on high doses of CytoD (Cofilin-1 localization at bridges, overgrowth of the ESCRT-III cone) were reproduced by using high doses of LatA, and the conclusions based on low doses of LatA (rescue of abscission in Cofilin-1-depleted cells) were reproduced by using low doses of CytoD. Therefore, the same results were obtained with the two drugs, even if they depolymerize actin through different mechanisms. This is now discussed in the context of the literature (**p. 12**). In particular, LatB is known to be less effective than LatA and is thought to be inactivated by serum (ref. 67). This likely explains why the abscission delay had not been found in ref. 13.

Note that for the sake of consistency, we now presented abscission experiments with CytoD in the main Figures and all LatA experiments in Supplementary Figures.

2) Low dose LatA is shown to restore abscission timing in cofilin-1-depleted cells, while the authors suggest “that the branched F-actin meshwork acts as a physical barrier that blocks the abnormal extension of the ESCRT-III polymers”. So, what about higher doses of LatA (of course added after midbody formation)? If the model is true, one would expect delayed MT-severing and delayed abscission, whether cofilin-1 is present or not.

As requested, we investigated the impact of adding high doses of LatA (100 nM) after bridge formation. Interestingly, we observed that both the MT cut and the membrane cut (abscission) were delayed (**new Fig. S5d**). As observed upon CK666 addition (Fig. 6d), the delay in abscission was fully explained by the MT cut delay (**new Fig. S5d**). We also found that these doses of LatA leads to an overgrowth of ESCRT-III cones (**new Fig. S6e**), as previously found after both CytoD (Fig. S6d) and CK666 treatments (Fig. 7d). Altogether, these new results back up our proposed model that the branched F-actin meshwork acts as a physical barrier that blocks the abnormal extension of the ESCRT-III polymers, and thereby promotes the MT cut.

Despite many tests, we could not find a concentration of LatA that alone could depolymerize enough F-actin to affect abscission and would not strongly perturb the Cofilin-1-depleted cells. Indeed, the combination of high doses of LatA and depletion of Cofilin-1 was very harmful for the cells (this led to extensive blebbing, rapid cell shape changes and increased cell death) and could not allow us to follow them for extended period of time necessary for measuring abscission timing by videomicroscopy.

To further strengthen the fact that F-actin is functionally important for the timing of abscission, we now presented results showing that depolymerizing actin with high doses of CytoD (0.4 μ M) after furrow ingression delays abscission by delaying the MT cut (**new Fig. S5c**). Thus, depolymerizing actin using CytoD or LatA led to similar results on ESCRT-III cone elongation and MT cut delay.

3) On the subject of F-actin drugs, I cannot help but wonder what the effects of low doses of the F-actin stabilising drug, jasplakinolide (or SiR-actin, which is essentially jasplakinolide), are on MT-severing, abscission timing, GFP-ARPC4 localization etc.. in control and cofilin-1-depleted cells. I do not feel this is a requirement for publication, but it could certainly provide further mechanistic insight and strengthen the model. Have the authors looked at this at all?

We tested the effect of stabilizing F-actin by Jaspakinolide (Jasp, 50 nM) and found interesting results. The combination of Jasp and Cofilin-1 depletion was detrimental for the cells and was not investigated further. Similarly to what we previously found upon Cofilin-1 depletion (main Fig. 3d and 4b), Jasp addition both delayed MT severing (**new Fig. S4b**) and increased the residence time of GFP-ARPC4 at the secondary ingression (**new Fig. S3g**). In addition, we found that Jasp reduces the growth rate of the ESCRT-III cone (**new Fig. S4c**). Importantly, the same results were observed in Cofilin-1-depleted cells (**new main Fig. 5d**).

Altogether, we conclude that stabilizing F-actin and perturbing F-actin remodeling by depleting Cofilin-1 led to very similar defects. This suggests that Cofilin-1 not only controls

the residence time and disappearance of the Arp2/3 pool at the secondary ingression but also promotes actin remodeling to favor the extension of the ESCRT-III cone. We have now added in the discussion (**p. 14**) that similar to Jasp, "Cofilin-1 depletion might thus reduce the dynamics of the branched F-actin pool at the tip of the ESCRT-III cone. We speculate that in Cofilin-1-depleted cells, the presence of a non-dynamic pool of F-actin at this location creates local resistance that slows down the elongation of the ESCRT-III cone and prevents normal CHMP1B incorporation."

We also noticed that once the MT are cut, Jasp also delays the membrane cut (**new Fig. S4b, b>a**), which is not observed in Cofilin-1-depleted cells. This suggests that F-actin depolymerization or increased F-actin dynamics (inhibited by Jasp) also plays a role later for the membrane cut independently to Cofilin-1.

4) siCofilin-1 extended the residence time of GFP-ARPC4, but did this result in enhanced GFP-ARPC4 accumulation above WT levels? This looks to be the case from the examples shown in Fig. 3d, but is this true at the population level? Or was there a fixed quantity of ARPC4 recruited, which simply persisted for longer?

We analyzed the levels of GFP-ARPC4 (maximum levels) in our movies and did not observe a significant change in Cofilin-1-depleted cells, compared to control cells (**new Fig. S3j**). Note that there is a broad range of GFP-ARPC4 intensities both in control and in Cofilin-1-depleted cells, and that the presented snapshots in Fig. 3d reflects this variability.

To confirm this result, we stained for endogenous ARPC2 and phalloidin, and found that there was no change of ARPC2 nor F-actin levels at the secondary ingression in Cofilin-1-depleted cells (**new Fig. S3i and S3j**).

Given that the growth rate of the ESCRT-III cone is slowed down in Cofilin-1-depleted cells (see point 3) above), we suspect that the dynamics of the actin filaments themselves within this actin pool matters. The fact that the F-actin stabilizing drug Jasplakinolide phenocopies Cofilin-1 depletion strongly suggests that indeed proper actin dynamics in this actin pool at the tip of the cone is important for ESCRT-III polymerization toward the MT cut site (**discussion, p 14.**).

5) Fig. 2 After TMP removal (5 min or 120 min), sequestration of Cofilin-1-GFP at the mitochondria is clearly shown, and this is an exciting and elegant approach. However, it would strengthen the data to show and quantify the presumed depletion at the intercellular bridge, which is the relevant pool for the accompanying data on abscission timing.

As requested, we now show Cofilin-1-GFP depletion in the ICB upon TMP removal in **new Fig. S2d**. The quantification of the mean intensity of Cofilin-1-GFP indicates a strong decrease of Cofilin-1-GFP at the secondary ingression upon TMP removal (**new Fig. S2d**). Thus, this method reduces Cofilin-1-GFP at the secondary ingression in an efficient and acute manner.

Also, in Fig. 2e: when exactly was TMP removed? It would be helpful to label it on the graph and specify in the legend.

Our aim is to trap Cofilin-1-GFP after furrow ingression. As indicated in detail in the method section (p. 18) and in the figure legend, TMP was thus removed one hour after replating metaphase cells on collagen —at that time, most cells have formed their bridge and cells that did not were excluded from the analysis. Then, we started image acquisition. For more clarity, we added “After furrow ingression:” on the graph in Fig. 2e.

6) The abstract lacks any methodology. I understand space is tight, but at the very least the model system should be mentioned.

We added "in HeLa cells" in the first sentence describing the results.

7) Also in the Abstract: “Mechanistically, branched F-actin functions as a physical barrier that limits ESCRT-III cone elongation and thereby favors stable spastin recruitment.” This sentence is too speculative for the abstract and should be restricted to the Discussion.

To make it clear that this is our interpretation, the sentence now reads, after the description of the results: " Mechanistically, we propose that branched F-actin functions as a physical barrier that limits ESCRT-III cone elongation and thereby favors stable spastin recruitment".

Typos

p4 “non-phosphorylable Cofilin-1” should be non-phosphorylatable Cofilin-1
p8 “significatively” should be “significantly”

Thanks for spotting these typos. These have been corrected.

Reviewer #2

The manuscript submitted by Advedissian, Fremont and Echard highlight a role for the actin binding proteins Cofilin-1 and ARPC4 in the mechanism of cytokinetic abscission. This is a well written manuscript that provides several new important insights to the process of abscission. First, they show that microtubule severing and membrane cutting are temporally separated, as previously suggested from in vivo studies. Second, they reveal a role for cofilin-1 in late cytokinesis and demonstrate that depolymerization of actin at late abscission stages can rescue the abscission delay observed in cell depleted of Cofilin-1. Last, and perhaps most striking are their findings that the Arp2/3 protein ARPC4 has a role in positioning the ESCRT-III filament at the site of microtubules and membrane scission suggesting that microtubule, actin and ESCRT act in concert to determine the site of abscission. Overall, I find this work significant and suitable for publication in Nature communications. I have several comments that should be addressed prior to publication.

Specific comments:

1. The data presented in Fig. 1 indicate that cofilin levels peaks ~15 minutes prior to MT cut. Yet, in figure 3c the peak is ~5 minutes prior to MT cut. Can the authors please explain these differences?

We now provide the distribution of the peak times of Cofilin-1 with respect to the MT cut. As shown in **new Fig. S3b**, approximately half of the cells display a peak of Cofilin-1 (and ARPC4) 5 min prior to the MT cut, which is why the peak is at 5 min in the curve representing the mean value (Fig. 3c). However, approximately 20% of the cells show a peak earlier (see also error bars in Fig. 3c), 15 min prior to the MT cut, as shown in the Fig. 1 of the previous version of the manuscript. To show the most frequent case, we provide in **new Fig. 1c** an example with Cofilin-1 that peaks 5 min before the MT cut. We also added that the maximal recruitment of Cofilin-1 and Arp2/3 occurred 5 min before the MT cut "in most cells" (**p. 7**).

2. I am having a hard time consolidating the cofilin-1 and ARPC4 findings. While I find the data presented for ARPC4 and the model that branched actin determine the site of ESCRT-III positioning, spastin recruitment and ultimately MT severing and abscission very convincing it is not clear to me how cofilin fit to this scenario. I guess the data support an upstream effect for cofilin. However, depolymerization of the actin pool at the constriction site by cofilin would suggest a downstream effect. This point should be clarified either by performing additional experiments to reveal the relationships between cofilin and ARPC4 at the intercellular bridge or by providing a reasonable explanation in the discussion that is well aligned with the data presented and with the known function of these proteins.

Our new results indicate that Cofilin-1 plays two successive roles when it is recruited at the tip of the ESCRT-III cone together with Arp2/3.

First, we now report that Cofilin-1 depletion reduces the growth rate of the ESCRT-III cone, indicating an upstream (early) role of Cofilin-1 (**new main Fig. 5d**). This is likely linked to a role of Cofilin-1 in regulating F-actin dynamics/turnover during cone elongation, since we observed the same results in non-depleted cells treated with the actin stabilizing drug Jasplakinolide (Jasp, 50 nM) (**new Fig. S4c**). Furthermore, Jasp addition after bridge formation delayed MT severing (**new Fig. S4b**), as previously found upon Cofilin-1 depletion (main Fig. 4b). The fact that the levels of F-actin (or ARPC2) at the secondary ingression are not increased in Cofilin-1-depleted cells (see Reviewer #1 point #4) is consistent with a role of Cofilin-1 in regulating the local remodeling (rather than quantities) of branched F-actin at the tip of the ESCRT-III cone.

Second, as shown in the original version of the manuscript, Cofilin-1 depletion increases the residence time of Arp2/3 at the secondary ingression (Fig. 3d). As pointed out by the Reviewer, this indicates a downstream (late) role of Cofilin-1 to clear the branched actin pool at the time of the MT cut. We now show that Jasp addition after bridge formation increased the residence time of GFP-ARPC4 at the secondary ingression (**new Fig. S3g**), as found upon Cofilin-1 depletion. This points to a function of Cofilin-1 in promoting branched actin disassembly at the secondary ingression at the time of the MT cut.

Altogether, we conclude that stabilizing F-actin after bridge formation and perturbing F-actin remodeling by depleting Cofilin-1 led to very similar defects. This suggests that Cofilin-1 not only controls the residence time and disappearance of the Arp2/3 pool at the secondary ingression but also promotes actin remodeling to favor the extension of the ESCRT-III cone. We have now added in the discussion (**p. 14**) that similar to Jasp, "Cofilin-1 depletion might thus reduce the dynamics of the branched F-actin pool at the tip of the ESCRT-III cone. We speculate that in Cofilin-1-depleted cells, the presence of a non-dynamic pool of F-actin at this location creates local resistance that slows down the elongation of the ESCRT-III cone and prevents normal CHMP1B incorporation."

3. The authors show that cofilin and ARPC4 affect spastin recruitment. Based on these findings they conclude that these proteins function upstream to spastin. To substantiate this conclusion, the arrival and organization of these actine binding proteins to the intercellular bridge should be examined in cells depleted of spastin.

As requested, we depleted spastin (**new Fig. S4d**) and quantified the percentages of secondary ingressions positive for endogenous Cofilin-1 and endogenous ARPC2. We observed that:

- the depletion of spastin did not impact on Cofilin-1 recruitment at the tip of the ESCRT-III cone (**new Fig. S4d**)
- the depletion of spastin did not prevent the recruitment of ARPC2 at the tip of the ESCRT-III cone (**new Fig. S6b**).

Altogether, these results are consistent with the idea that Cofilin-1 and Arp2/3 function upstream to spastin.

4. Few of the intercellular bridges shown in the figures has abnormal morphologies or atypical CHMP4B organization. Fig 3d, microtubule staining looks abnormal. Fig 6b 20 min, CHMP4B staining looks abnormal, Fig 7 bridge microtubules in cells treated with CK666 appear to fade over time suggesting continuous depolymerization. If these images reflect a abnormalities resulting from the manipulations the cells were subjected to, it should be clearly stated in the text. Otherwise, these images should be replaced with more representative examples.

We comment below each figure highlighted by Reviewer 2:

- Fig. 3d (Cofilin-1 depletion): the snapshots show a bent bridge that is purposely focused on the midbody and the secondary ingression. The remaining bridge MTs are present on lower planes and thus appear blurry or out of focus, especially in the first time points (the ICB often move in the Z axis over 30 min periods). We have not observed any defects in MTs upon Cofilin-1 depletion either in fixed cells or in movies.

- Fig. 6b: CHMP4B is initially present in two cones on each side of the midbody (time points 10 and 15 min). At the time point 20 min, the cones dissociate from the midbody pool, leading to a discontinuous/interrupted appearance of CHMP4B-GFP. This is often seen in time-lapse imaging of CHMP4B-GFP (as in this video) and for endogenous CHMP4B in fixed samples (see examples 1 and 2 below). This dissociation and fragmented staining have also

been documented for IST1 and is not abnormal in HeLa cells (Goliand *et al.* Cell Reports 2018, ref. 15).

Legend: Staining of endogenous CHMP4B (green) and Tubulin (magenta) in two different intercellular bridges in HeLa cells. Bracket: midbody (MB). Arrowhead: secondary ingression. Scale bars = 5 μm .

- Fig. 7: This video is representative of what happens upon CK666 treatment. We agree with Reviewer 2 that the MTs appear to fade away, suggesting continuous depolymerization upon CK666 treatment. Interestingly, this is also observed upon spastin depletion (ref. 13), reinforcing the notion of an abnormal spastin-mediated severing in CK666-treated cells. We now discuss this point (**p. 14**).

5. It was previously shown that the actin/tubulin cross linking protein Gas2l3 localize to the construction site of the intercellular bridge (Pe'er *et al.* PLOS One, 2013). It will be really interesting to examine whether this protein plays a role in the ARPC4 dependent spastin and ESCRT-III localization shown here. At the minimum this paper should be referred to in the discussion chapter.

To investigate this possibility, we co-stained intercellular bridges with antibodies against endogenous ARPC2 and endogenous Gas2l3. We confirmed that Gas2l3 and ARPC2 are enriched at the midbody side, but never at the same time. Indeed, we could not find bridges positive for both Gas2l3 and ARPC2 (see figure below). Based on the degree of pinching of the tubulin staining, our pictures suggest that Gas2l3 is removed before ARPC2 appears.

The lack of co-localization between Gas2l3 and ARPC2 at the secondary ingression suggests that there is not a direct crosstalk between these proteins and we thus did not investigate further the role of Gas2l3 in the Arp2/3-dependent spastin and ESCRT-III localization. However, we cannot exclude that Gas2l3 might play a functional role and, as suggested by the Reviewer, we added this idea in the discussion (**p. 13**): "Further work would be needed

to investigate whether actin and tubulin crosslinkers known to localize at the secondary ingression, such as the protein Gas2l3^{68,69}, might regulate the assembly of the branched actin pool at this location and thereby promote the MT cut."

a

Legend: Staining of endogenous Gas2l3 (green), Tubulin (blue) and ARPC2 (magenta) in intercellular bridges in HeLa cells. In these two examples, the same acquisition parameters and contrasts were applied for the Gas2l3 and ARPC2 channels. Bracket: midbody (MB). Arrowhead: secondary ingression/pinch of Tubulin staining. Scale bars = 5 μm .

Minor comments:

1. Because abscission timing was measured based on phase contrast and/or microtubule labeling it is important to specify on each plot what was measured to avoid confusion. This is relevant to all plots in Fig 2, Fig 4d, Fig 6 c, e.

This has been now indicated on the y-axis in each graph where we precise that "abscission" means "membrane cut" (Figures 2b, 2c, 2e 2f, 4b, 4d, 6c, 6d, 6e, S2e, S4a, S4b, S5c, S5d). We also specify that the abscission is measured by phase contrast and that the MT cut is based on the SiR-tubulin channel in the corresponding Figure legends.

2. Titles in Figures 3, 4, 5 should be toned down to reflect the data presented in each figure and avoid over interpretation.

As requested, the title of Figure 3 now reads "Cofilin-1 colocalizes with a pool of branched F-actin at the secondary ingression before the MT cut and regulates the lifetime of this pool." to remove any interpretation of the results presented in this Figure.

We think that the title of Figure 4 appropriately reflects the data presented in this Figure. In our opinion, the fact that 1) Cofilin-1 depletion delays the MT cut but not the membrane cut afterwards, and 2) the experimental depolymerization of MTs in Cofilin-1-depleted cells fully rescues normal abscission shows that "Cofilin-1 promotes abscission by favoring MT severing

but does not influence the membrane cut occurring later." We would be happy to discuss it further if this Reviewer feels that it is necessary.

The title of Figure 5 now reads "The MT severing delay observed in Cofilin-1-depleted cells is associated with a defective recruitment of the ESCRT-III subunit CHMP1B and spastin." to remove any interpretation of the results presented in this Figure.

3. The abstract statement: "contrary to the common assumption" is a bit misleading. It is correct that the field generally referred to microtubule severing as an indicative for abscission, but the notion that membrane severing itself occurs later is not new to the field and has been previously discussed. Please remove this statement from the abstract or tone it down considerably.

As requested, we replaced the initial sentence by "Here, we found that the microtubule and the membrane cuts are two separate events that are regulated differently." and removed "contrary to the common assumption".

REVIEWERS' COMMENTS

Reviewer #1 (Remarks to the Author):

The authors have more than satisfactorily responded to my initial comments and concerns. The changes have strengthened the conclusions and I recommend publication of this very interesting study.

Reviewer #2 (Remarks to the Author):

The authors have fully addressed my comments. I recommend publication.